# `Distilled Decoding 2`: One-step Sampling of Image Auto-regressive Models with Conditional Score Distillation

**Enshu Liu**
Tsinghua University
Beijing, China

**Qian Chen**
Tsinghua University
Beijing, China

**Xuefei Ning**
Tsinghua University
Beijing, China

**Shengen Yan**
Infinigence-AI
Beijing, China

**Guohao Dai**
Shanghai Jiaotong University
Shanghai, China

**Zinan Lin**[*][†]
Microsoft Research
Redmond, WA, USA

**Yu Wang**[†]
Tsinghua University
Beijing, China

## Abstract

Image Auto-regressive (AR) models have emerged as a powerful paradigm of visual generative models. Despite their promising performance, they suffer from slow generation speed due to the large number of sampling steps required. Although Distilled Decoding 1 (DD1) was recently proposed to enable few-step sampling for image AR models, it still incurs significant performance degradation in the one-step setting, and relies on a pre-defined mapping that limits its flexibility. In this work, we propose a new method, `Distilled Decoding 2` (DD2), to further advance the feasibility of one-step sampling for image AR models. Unlike DD1, DD2 does not without rely on a pre-defined mapping. We view the original AR model as a teacher model that provides the ground truth conditional score in the latent embedding space at each token position. Based on this, we propose a novel *conditional score distillation loss* to train a one-step generator. Specifically, we train a separate network to predict the conditional score of the generated distribution and apply score distillation at every token position conditioned on previous tokens. Experimental results show that DD2 enables one-step sampling for image AR models with a minimal FID increase from 3.40 to 5.43 and 4.11 to 7.58 on ImageNet-256, while achieving $8.0\times$ and $238\times$ speedup with VAR and LlamaGen models, respectively. Compared to the strongest baseline DD1, DD2 reduces the gap between the one-step sampling and original AR model by 67%, with up to $12.3\times$ training speed-up simultaneously. DD2 takes a significant step toward the goal of one-step AR generation, opening up new possibilities for fast and high-quality AR modeling. Code is available at https://github.com/imagination-research/Distilled-Decoding-2.

## 1  Introduction

Image autoregressive (AR) models have recently achieved state-of-the-art performance in high-fidelity image synthesis, surpassing other generative approaches such as VAEs, GANs, and diffusion models [38, 3, 6, 27, 15, 45, 2, 16, 37, 35, 17, 9, 14, 8, 31, 10, 34].

---

[*]Project Advisor: Zinan Lin.

[†]Correspondence to Zinan Lin (zinanlin@microsoft.com) and Yu Wang (yu-wang@mail.tsinghua.edu.cn).

39th Conference on Neural Information Processing Systems (NeurIPS 2025).

Despite their strong generation ability, a key limitation of AR models lies in their inherent sequentially modeling manner, which leads to the token-by-token sampling process and significantly slower inference speed. Numerous methods have been proposed to reduce sampling steps [41, 1, 18, 36, 11, 2, 37, 17], but nearly all fail to achieve single-step sampling without significant performance degradation, leaving room for further speedup. Please refer to Sec. 4.1 for more details.

**Distilled Decoding 1** (**DD1**) [21] marks a significant breakthrough in reducing the sampling steps for AR models, as it is the first method capable of compressing the sampling process of an AR image model to *only a single step*. DD1 introduces flow matching [22, 20] into the AR sampling pipeline. Specifically, instead of sampling the next token directly from a probability vector output by the AR model, DD1 leverages flow matching in the codebook embedding space to transform a *noise token* into a *data token*. This enables token-wise deterministic mapping from noise to data while preserving the output distribution of the original AR model. By iteratively conducting this process following the original AR sampling

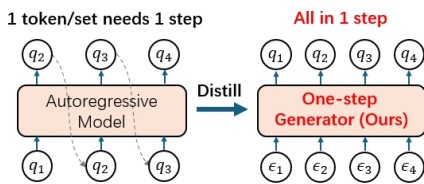

Figure 1: Our goal is to distill a multi-step AR model in to a one-step generator while keeping its distribution.

order, DD1 obtains a complete mapping from a noise token sequence to a data token sequence. Then, a new model is distilled to directly learn this mapping, allowing the generation of the entire token sequence in a single forward pass.

However, the constructed mapping is inherently challenging for the model to learn, resulting in a noticeable performance drop compared to the original AR model. In addition, training a generative model to directly fit a predefined mapping may impose constraints on the flexibility. In contrast, models like GANs and VAEs, which do not learn explicit input–output correspondences, have shown broad applicability across downstream generation tasks [19]. This insight leads us to ask:

*Can we train a one-step generative model whose output distribution matches a given AR model, without relying on any predefined mapping?*

To answer this problem, we propose `Distilled Decoding 2` (**DD2**) as a completely new method. Inspired by DD1, our key motivation is to reinterpret the AR model, which originally outputs a discrete probability vector for the next token $q_i$, as a *conditional score model* that predicts the gradient of the log conditional probability density (i.e., the conditional score) in the codebook embedding space. Specifically, we view the generation of each token as a conditional flow matching process. Based on this, given all previous tokens $q_{1,...,i-1}$ as the condition, we can use the teacher model's output probability vector to define a conditional score $s(q_i^t, t|q_{<i}) = \nabla_{q_i^t} \log p(q_i^t|q_{<i})$, where $t$ denotes the flow matching timestep. Unlike DD1, where the conditional score is used solely to construct an ODE-based mapping, we aim to make fuller use of this signal. We borrow ideas from *score distillation methods* (e.g., [26, 40, 24, 43, 48]), which match the score of ax one-step generator's distribution to that of a teacher diffusion model and have recently shown strong performance in diffusion-based generation. Specifically, our DD2 jointly trains a **one-step generator** and a **conditional guidance network** that learns the conditional score of the generator distribution. We propose a novel *Conditional Score Distillation (CSD)* loss for training, which aligns the conditional score between the guidance network and the teacher AR model at every token position. We show that when the CSD loss is minimized to its optimality, the output distribution of the one-step generator matches exactly that of the original AR model.

It is important to highlight that our method is **fundamentally different** from *diffusion* score distillation. Although both approaches involve aligning scores, AR models and diffusion models follow completely different modeling approaches and generation processes. As a result, the goals and challenges in this paper are inherently distinct from previous works. More discussion about the differences between the two methods can be found at Sec. 6.1.

To validate the effectiveness of DD2, we follow the evaluation setup of DD1 and conduct experiments on ImageNet-256 [4] with two strong autoregressive models: VAR [37] and LlamaGen [35]. On VAR, we reduce the sampling steps from 10 to 1 with a marginal FID increase less than 2.5 (e.g., from 4.19 to 6.21), achieving a up to 8.1× speedup. Compared to DD1, DD2 reduce the performance gap between the 1-step model and the original AR model by up to 67%. On LlamaGen, we compress the sampling process from 256 steps to 1 with an FID degradation from 4.11 to 8.59, resulting in a 238× speedup. Compared to DD1, our 1-step model achieves an FID improvement of 2.76. Further

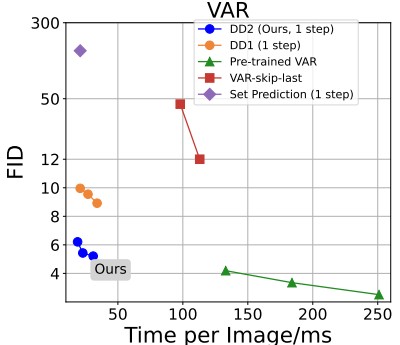 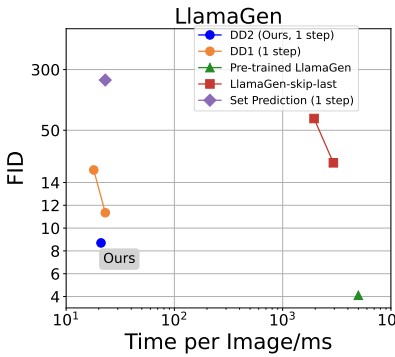

Figure 2: Comparison of DD2 models, DD1 models, pre-trained models, and other acceleration methods for pre-trained models. DD2 achieves a significant speedup compared to pre-trained models while outperforming DD1 by a large margin. Other methods fail to achieve one-step sampling. For DD2, DD1, and the pre-trained model, each point corresponds to a different model size, whereas for the skip-last method, each point corresponds to a different number of skipped final steps.

comparisons between DD2 and other baseline methods are presented in Fig. 2. Additionally, DD2 is highly efficient to train: compared to DD1, it achieves up to $12.3\times$ training speedup. We hope this work can inspire future research toward making image AR models maximally efficient while keeping their superior sample quality.

## 2    Preliminary

In this section, we introduce the formulation of standard image AR models to understand DD2.

### 2.1    Image Tokenizer

To train an image AR model, we first need to convert continuous-valued images into discrete token sequences, so that the probability of each token can be explicitly outputted by the model. Recent AR models mostly rely on vector quantization (VQ) [39], which leverages an encoder $\mathcal{E}$, a quantizer $\mathcal{Q}$, and a decoder $\mathcal{D}$ to discretize and reconstruct visual content.

The process begins by encoding the input image $x \in \mathbb{R}^{3 \times H \times W}$ into a latent representation: $z' = \mathcal{E}(x)$, where $z' = (z_1, z_2, \ldots, z_{h \times w}) \in \mathbb{R}^{C \times h \times w}$ is a lower-resolution feature map containing $h \times w$ embeddings, each of dimension $C$. For each embedding $z_i$, the quantizer selects the nearest code vector $q_i$ from a learned codebook $\mathcal{V} = (c_1, c_2, \ldots, c_V) \in \mathbb{R}^{V \times C}$. The resulting discrete token sequence is denoted as $z = (q_1, q_2, \ldots, q_{h \times w})$, where each $q_i$ is a token $c_j$ in $\mathcal{V}$. To reconstruct the original image, the decoder $\mathcal{D}$ takes $z$ as input and produces: $\hat{x} = \mathcal{D}(z)$. During training, a reconstruction loss $l(\hat{x}, x)$ is used to ensure fidelity between the original and the reconstructed image. This VQ-based framework underpins many state-of-the-art image AR models [16, 2, 37, 35, 17].

### 2.2    Auto-regressive Modeling

Once a well trained image tokenizer is available, an AR model can be employed for image generation. We assume that an image is represented as a sequence of discrete tokens $z = (q_1, \cdots, q_n)$, where each $q_i$ corresponds to an embedding from the codebook $\mathcal{V}$: $q_i \in \{c_1, \ldots, c_V\}$. The AR model is trained to estimate the conditional probability distribution of each token given all previous tokens: $p(q_i|q_{<i}) = p(q_i|q_{i-1}, q_{i-2}, \cdots, q_1) = (p_1, \ldots, p_j, \ldots, p_V)$, where $p_j$ denotes the probability that the next token corresponds to the $j$-th entry in the codebook.

At generation time, the model samples tokens one by one in order, and the likelihood of the full sequence is given by: $p(Z) = \prod_{i=1}^{n} p(q_i|q_{<i})$. This generation procedure requires $n$ autoregressive steps, which is often a large number, resulting in slow inference speed and limited efficiency.

## 3    Distilled Decoding 2

In this section, we first introduce the formal problem definition in Sec. 3.1. Then, we propose **Conditional Score Distillation (CSD)** loss as the core component of DD2 in Sec. 3.2. Then we discuss our initialization method in Sec. 3.3, which plays a crucial role in training speed and performance. Finally, we present the full training pipeline of our approach.

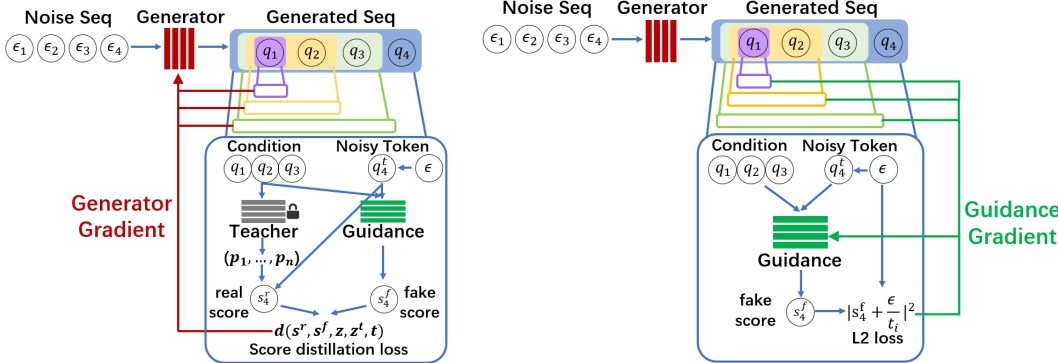

(a) Training loss of the generator.  (b) Training loss of the guidance network.

Figure 3: Training process using CSD loss. For the generator, the teacher AR model and the guidance network give the true and fake conditional score of *each noisy token* based on all previous clean tokens, respectively. Then the true and fake conditional score are used to calculate the score distillation loss, which produces gradient to train the generator. The guidance network learn the conditional score of *each noisy token* given all previous clean tokens, by optimizing with a standard AR-diffusion loss [17]. The generator and the guidance model are trained alternately.

### 3.1 Problem Formulation

Suppose a image can be encoded as a sequence of length $n$, and we have a well trained teacher AR model $p_\Phi$, which gives the next token probability conditioned on all previous tokens $p_\Phi(x_i|x_{<i})$ and will be fixed. Our goal is to train a one-step generator $G_\theta$, which can output a generated sequence $z_\theta = (q_1, \ldots, q_n)$ in one run given a latent variable $\varepsilon$ drawn from the prior distribution: $z_\theta = G_\theta(\varepsilon)$. We hope the distribution of $z_\theta$ can match the distribution of the teacher AR model.

### 3.2 Conditional Score Distillation Loss

In this section, we first introduce our idea of viewing teacher AR model as a conditional score model in Sec. 3.2.1, and propose the objective based on it in Sec. 3.2.2. Then, we present how to train the conditional score model for the generator distribution in Sec. 3.2.3, as it is required by the objective.

#### 3.2.1 Teacher AR as a Conditional Score Model

Considering the generation process of the $i$-th token $q_i$ given all previous tokens $(q_1, \ldots, q_{i-1})$ as the condition, we have the probability vector $p = (p_1, \ldots, p_V)$ outputted by the teacher AR model, where $p_j \geq 0$ denote the probability of $j$-th token $c_j$ and $\sum_{j=1}^{V} p_j = 1$. Inspired by DD1 [21], we view the sampling process as a continuous transformation of flow matching [22, 20] from a source Gaussian distribution at $t = 1$ to a sum of Dirac function $\delta(\cdot)$ weighted by $p$ at $t = 0$: $p(q_i) = \sum_{j=1}^{V} p_j \delta(q_i - c_j)$. By choosing the noise schedule of RectFlow [22], the score function can be expressed in closed form as:

$$s(x_t, t, p) = -\frac{\sum_{j=1}^{V} p_j(x_t - (1-t)c_j)e^{-\frac{(x_t - (1-t)c_j)^2}{2t^2}}}{t^2 \sum_{j=1}^{V} p_j e^{-\frac{(x_t - (1-t)c_j)^2}{2t^2}}} \tag{1}$$

For more details on the derivation of this expression, refer to App. C.1. By substituting $q_i$ as $x$ and $(p_1, \ldots, p_V) = p_\Phi(q_i|q_{<i})$ to Eq. (1), we rewrite the left side of Eq. (1) in the form of *conditional score function* $s(x_t, t, p) = s_\Phi(q_i^t, t|q_{<i})$. Here $q_i^t$ is a noisy version of the clean token $q_i$: $q_i^t = (1-t)q_i + t\epsilon, \epsilon \sim \mathcal{N}(0; \mathbf{I})$. The term *conditional score* refers to the score of $q_i^t$ conditioned on $q_{<i}$. Note that the condition term $q_{<i}$ consists of previous clean tokens without any noise injection, so it can also be noted as $q_{<i}^0$.

#### 3.2.2 Training Objective

With access to the true score function, we aim to make full use of this information rather than using it merely to construct an ODE mapping as in DD1. To this end, we draw inspiration from score distillation methods. These methods seek to align the distribution generated by a model with that of a teacher by matching their respective score functions.

We first present a general formulation of score distillation. Let $x \in \mathbb{R}^C$ be a random variable. Denote $p_\Phi$ and $s_\Phi$ as the *probability density function* and its *score function* given by the teacher model $\Phi$, $p_\theta$ and $s_{fake}$ as the *probability density function* and its *score function* of the generator $\theta$. A general score distillation loss can be given as:

$$\mathcal{L}_{SD} = \mathbb{E}_{t\sim[0,T],x_0\sim p_\theta,\epsilon\sim\mathcal{N}(0;\mathbf{I})} d(s_\Phi(\alpha_t x_0 + \sigma_t\epsilon, t), s_{fake}(\alpha_t x_0 + \sigma_t\epsilon, t)), \qquad (2)$$

where $d$ is a function satisfying that minimal $\mathcal{L}_{SD}$ guarantees $\forall x \in \mathbb{R}^C, p_\theta(x) = p_\Phi(x)$. In practice, we choose SiD loss [48] due to its effectiveness, giving:

$$d = \omega(t)\frac{\sigma_t^4}{\alpha_t^2}(s_\Phi - s_{fake})^T(s_\Phi + \frac{\epsilon}{\sigma_t} - \alpha(s_\Phi - s_{fake})), \qquad (3)$$

where $\omega(t)$ is the weight function and $\alpha$ is a hyper-parameter, which we set to 1.0.

In our scenario, however, we are not aligning the distribution of a single random variable like score distillation for diffusion models [43, 24, 48], but a sequence of random variables with auto-regressive correspondence. Specifically, we aim to match the generator's conditional distribution at each token position with that of the teacher AR model. This motivates us to minimize the score distillation loss on all token positions. Additionally, we have to replace the score term in Eq. (2) with the conditional score given all previous tokens and $\alpha_t, \sigma_t$ with the noise schedule of RectFlow [22]. By incorporating above modifications to Eq. (2), we propose our conditional score distillation (CSD) loss:

$$\mathcal{L}_{CSD} = \mathbb{E}_{t_i,(q_1,...,q_n)\sim p_\theta,\epsilon\sim\mathcal{N}(0;\mathbf{I})} \sum_{i=1}^{n} d(s_\Phi(q_i^{t_i}, t_i|sg(q_{<i})), s_{fake}(q_i^{t_i}, t_i|sg(q_{<i}))), \qquad (4)$$

where $q_i^t = (1-t)q_i + t\epsilon$ and $sg(\cdot)$ means the stop gradient operation. We give the following proposition to show the correctness of our CSD loss, with a brief proof in App. A.

**Proposition 1.** *Minimal $\mathcal{L}_{CSD}$ guarantees $\forall z = (q_1, \ldots, q_n) \in \mathbb{R}^{n*C}, p_\theta(z) = p_\Phi(z)$.*

Intuitively, Eq. (4) encourages progressive alignment of the token sequence distributions. Consider the first token $q_1$, which has no constraints by any other tokens. Its associated loss term reduces to a standard score distillation loss $\mathbb{E}_{t_1,q_1\sim p_\theta,\epsilon\sim\mathcal{N}(0;\mathbf{I})} d(s_\Phi(q_1^{t_1}, t_1), s_{fake}(q_1^{t_1}, t_1))$, which encourages $p_\theta(q_1)$ to align with $p_\Phi(q_1)$. Once the first token's distribution is aligned, we then consider the loss for the second token: $\mathbb{E}_{t_2,q_2\sim p_\theta,\epsilon\sim\mathcal{N}(0;\mathbf{I})} d(s_\Phi(q_2^{t_2}, t_2|sg(q_1)), s_{fake}(q_2^{t_2}, t_2)|sg(q_1))$. Optimizing this ensures $p_\theta(q_2|q_1) = p_\Phi(q_2|q_1)$. Given that $p_\theta(q_1) = p_\Phi(q_1)$ has already been achieved, it follows $p_\theta(q_1, q_2) = p_\Phi(q_1, q_2)$. By sequentially matching the distribution on each token position, we can finally align the entire distribution $p_\theta(q_1, \ldots, q_n)$ with $p_\Phi(q_1, \ldots, q_n)$.

### 3.2.3 Learning the Conditional Score of the Generator

To optimize Eq. (4), we need to access the conditional score of the generator $s_{fake}(q_i^{t_i}, t_i|q_{<i})$. Following previous works of diffusion score distillation [24, 43, 42, 48], we train a separate model $\psi$ to output this term, which we refer to as the conditional guidance network.

Specifically, our guidance network consists of a decoder-only transformer backbone and a lightweight MLP head with negligible cost. The training procedure is inspired by MAR [17]. Given a generated token sequence $(q_1, \ldots, q_n)$ from the generator, we first process it with the causal transformer backbone, yielding a sequence of hidden features $(f_1, \ldots, f_n)$. Each feature $f_i$ only corresponds to tokens $q_{<i}$ and thus captures strictly causal context. For each token position $i$, the MLP takes as input a noised version of the token $q_i^{t_i}$, the corresponding timestep $t_i$, and the contextual feature $f_i$. Since $f_i$ only corresponds to $q_{<i}$ as the conditioning, we denote the outputted score function as the fake conditional score $s_\psi(q_i^{t_i}, t_i \mid q_{<i})$. We train the model across all AR positions in parallel and then present the following loss:

$$\mathcal{L}_{FCS} = \mathbb{E}_{t_i,(q_1,...,q_n)\sim p_\theta,\epsilon\sim\mathcal{N}(0;\mathbf{I})} \sum_{i=1}^{n} \|s_\psi(q_i^{t_i}, t_i|q_{<i}) - \nabla_{q_i^{t_i}} \log p(q_i^{t_i}|q_i)\|^2, \qquad (5)$$

where $q_i^t = (1-t)q_i + t\epsilon$, and $\nabla_{q_i^{t_i}} \log p(q_i^{t_i}|q_i)$ can be simplified to $-\frac{\epsilon}{t}$ [34]. The MLP and transformer backbone are jointly optimized with $\mathcal{L}_{FCS}$.

In practice, the guidance network $\psi$ and generator $\theta$ are trained alternately using Eq. (5) and Eq. (4), respectively. During generator training with Eq. (4), the score term $s_{fake}$ is entirely replaced by $s_\psi$, with gradients blocked from propagating into $\psi$. The training algorithm and an illustration of the pipeline are provided in Alg. 1 and Fig. 3.

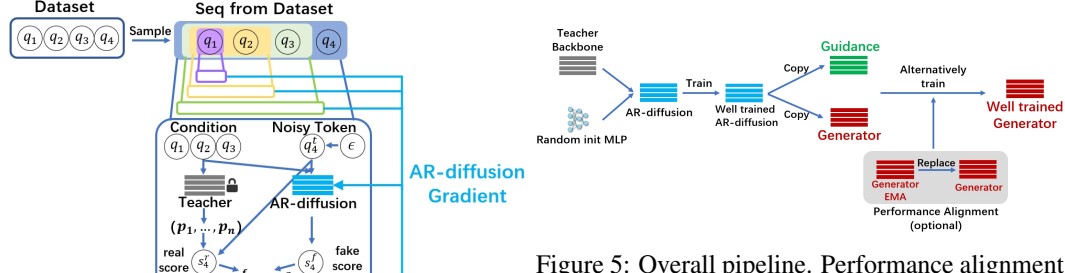

Figure 4: Training loss for initialization.

Figure 5: Overall pipeline. Performance alignment is an optional technique in the CSD training stage, which is introduced in App. B.4.

## 3.3 Initialization of Generator and Guidance Network

With the training procedure outlined in Alg. 1, we are now ready for DD2 training. However, directly applying this method does not yield satisfactory results. We attribute this to the poor model initialization. We delve into this issue and propose our solutions in the following part of this section.

We find that good initialization is crucial for score distillation methods: poor initialization can lead to slow convergence or even training collapse. To validate this, we conduct *diffusion distillation* experiments on the ImageNet-64 dataset using the original DMD [43, 42] approach under different initialization schemes: **(1) Default**: both the guidance and generator models are initialized from a pretrained teacher diffusion model, **(2) Random Guidance**: the guidance model is randomly initialized, **(3) Random Generator**: the generator is randomly initialized, and **(4) Partial Random Generator**: only the final layer of the generator is randomly initialized. As shown in Fig. 6, improper initialization of either the guidance or the generator leads to significant training degradation. Even randomly initializing just the final layer of the generator severely impacts the performance. This is because initialization determines both the internal knowledge stored in the network and the generator's initial distribution, both of which are critical to stable and efficient score distillation training as discussed in [46].

In our setting, both the generator and the conditional guidance network output continuous values, while the teacher AR model produces probability vectors. This structural mismatch makes it impossible to directly reuse the model weights from the teacher AR model to initialize the output heads. To address this, we propose a novel initialization strategy: we first replace the teacher AR model's classification head with a lightweight MLP, and fine-tune the new model with AR-diffusion loss [17] to align its distribution with the teacher AR model. This process is similar to the training of the conditional guidance network and MAR model [17] but with a key difference: we introduce Ground Truth Score (GTS) loss by replacing the Monte Carlo Estimation in Eq. (5) with the ground truth score calculated using the teacher AR model with Eq. (1), giving:

$$\mathcal{L}_{GTS} = \mathbb{E}_{t_i,(q_1,\dots,q_n)\sim p_\Phi,\epsilon\sim\mathcal{N}(0;\mathbf{I})} \sum_{i=1}^{n} \|s_\psi(q_i^{t_i},t_i|q_{<i}) - s_\Phi(q_i^{t_i},t_i|q_{<i})\|^2. \tag{6}$$

This loss significantly improves training stability and convergence speed, as demonstrated in experiments in Tab. 7. The training process is shown at Alg. 2

For both generator and guidance network, we adopt the same architecture composed of a transformer backbone and a lightweight MLP head, both initialized from the tuned AR diffusion model. For generator, we sample a noise sequence $\varepsilon = (\epsilon_1,\dots,\epsilon_n)$ as the latent variable input, where each $\epsilon_i \sim \mathcal{N}(0;\mathbf{I})$. This sequence is fed directly into the MLP, while a one-step offset version is provided to the transformer backbone. Model architectures are shown in Fig. 7.

Such strategy serves as a strong initialization for both generator and guidance network, improving the training significantly as demonstrated in Sec. 5.5.

**Overall pipeline** The complete training process consists of two stages: an initialization tuning phase with Eq. (6) and a main training phase with Eq. (4) (for the generator) and Eq. (5) (for the guidance network). The workflow is illustrated in Alg. 3 and Fig. 5. More techniques can be found in App. C.

**Multi-step sampling involving the teacher model.** To achieve more flexible trade-off between sample quality and steps, we can use the teacher model to refine the last several steps of the one-step generated sequence. Details of this sampling method are shown in App. D and Alg. 4.

| **Algorithm 1:** Training with CSD loss | **Algorithm 2:** Tuning the AR-diffusion model |
|---|---|
| **Require:**
    the pre-trained teacher AR model $\Phi$.
1: **while** not converged **do**
2:    $z = (q_1, \ldots, q_n) \leftarrow G_\theta(\epsilon)$

    // **Train generator** $\theta$
3:    Sample $t = (t_1, \ldots, t_n)$, sample $z_\epsilon = (\epsilon_1, \ldots, \epsilon_n)$ from Gaussian distribution
4:    $z^t = (q_1^{t_1}, \ldots, q_n^{t_n}) \leftarrow (1-t)z + tz_\epsilon$
5:    update $\theta$ with $\mathcal{L}_{CSD}(z^t, z, t, \psi, \Phi)$ // Eq. (4).

    // **Train guidance network** $\psi$
6:    Sample another $t = (t_1, \ldots, t_n)$, sample another $z_\epsilon = (\epsilon_1, \ldots, \epsilon_n)$ from Gaussian distribution
7:    $z^t = (q_1^{t_1}, \ldots, q_n^{t_n}) \leftarrow (1-t)z + tz_\epsilon$
8:    update $\psi$ with $\mathcal{L}_{FCS}(z^t, z, t, \psi)$ // Eq. (5).

9: **end while**
10: **return** $\theta$ | **Require:** dataset $\mathcal{D}$, the pre-trained teacher AR model $\Phi$, AR-diffusion model $\Psi$.
1: initialize the backbone of $\Psi$ with the backbone of $\Phi$, randomly initialize the MLP head of $\Psi$
2: **while** not converged **do**
3:    Sample $z = (q_1, \ldots, q_n) \sim \mathcal{D}$
4:    Sample $t = (t_1, \ldots, t_n)$, sample $z_\epsilon = (\epsilon_1, \ldots, \epsilon_n)$ from Gaussian distribution
5:    $z^t = (q_1^{t_1}, \ldots, q_n^{t_n}) \leftarrow (1-t)z + tz_\epsilon$
6:    update $\Psi$ with $\mathcal{L}_{GTS}(z^t, z, t, \Psi, \Phi)$ Eq. (6).
7: **end while**
8: **return** $\Psi$

**Algorithm 3:** Overall Pipeline

**Require:** dataset $\mathcal{D}$, the pre-trained teacher AR model $\theta^\Phi$.
1: train the AR-diffusion model $\Psi$ with Alg. 2
2: duplicate the trained $\Psi$ as generator $\theta$ and guidance network $\psi$
3: train generator $\theta$ and guidance network $\psi$ with Alg. 1
4: **return** $\theta$ |

## 4 Related Works

### 4.1 Reducing the Sampling Steps of AR Models

Many prior works have attempted to reduce the sampling steps of AR models. **Set prediction** is a commonly used approach in image AR modeling, where the model is trained to predict the probability of a set of tokens simultaneously [2, 16, 37, 17]. It significantly reduces the number of sampling steps to around 10. However, this method struggles to sample with very few steps (e.g., 1), due to the complete loss of token correlation within each set. As the set size increases, this loss becomes increasingly detrimental to sample quality as discussed in [21]. For example, consider the case where the dataset contains 2 data samples with 2 dimensions: $\mathcal{D} = \{(0,0), (1,1)\}$. The one-step sampling yields a uniform distribution among $\{(0,0), (1,1), (0,1), (1,0)\}$, which is incorrect. For more details, please refer to the Section 3.1 of DD1 [21]. **Speculative decoding** is another method of step reduction, which is widely used in large language models (LLMs) [41, 1, 18] due to its training-free property. It generates several draft tokens with a more efficient sampling method and then verifies them in parallel using the target model. Speculative decoding can achieve only a limited compression ratio of sampling steps (less than $3\times$) in image AR generation [36, 11], due to the weak modeling capacity of the draft generator.

Distilled Decoding 1 (DD1) [21] is the first work that compress the sampling steps of image AR models to 1 without performance collapse. The key idea of DD1 is to construct a deterministic mapping between a sequence of noise tokens and a sequence of target tokens. Specifically, given the probability vector outputted by the teacher AR model when generating the next token, DD1 replaces the multinomial sampling process of the original AR model with flow-matching sampling. The required velocity field can be accurately calculated through Eq. (1). By conducting this process following the original AR order, DD1 can map a noise sequence to a data sequence. Then DD1 simply train a neural network to fit this mapping. Although DD1 enables one-step sampling, it suffers from significant performance degradation and relatively slow training. Moreover, its reliance on a predefined mapping limits flexibility. As a new one-step training framework for AR models, DD2 effectively alleviates all the issues above.

### 4.2 Score Distillation for Diffusion Models

Similar to AR models, diffusion models (DMs) also suffer from a large number of sampling steps required by solving the diffusion ODE/SDE. Score distillation [26, 40] serves as a method to distill the multi-step teacher DM into a one-step generator [24, 43, 42, 48]. The main idea of score distillation is to make the distribution of the generator indistinguishable to the distribution of the teacher DM. A general formulation of score distillation generator loss has been given in Eq. (2). A guidance network is introduced to approximate the score function $s_{fake}$ of the generator. Both models are trained in turn. Different score distillation methods choose different types of generator loss. For example, [24, 43, 42] take the KL divergence between the two distributions as the objective, which gives $d = s_{fake} - s_\Phi$. SiD [48] aims to optimize the $l_2$ distance between the score functions of the two distributions, giving Eq. (3) as the loss. Compared to traditional diffusion distillation method

based on ODE mapping [23, 29, 33, 13, 32], score distillation methods are more flexible and have better results. By introducing training data, [46] eliminates the teacher guidance in score distillation through class-ratio estimation. Recently, there are also methods applying DMD to temporal causal data type like video [44]. The main difference between our paper and [44] lies in the problems they aim to solve: our work focuses on few-step sampling for AR models, where an AR model is available as the teacher, while [44] targets to decrease the step of DMs and assumes access to a teacher DM.

# 5 Experiments

In this section, we apply DD2 to existing pretrained AR models to demonstrate DD2's strong ability to compress AR sampling into a single step.

## 5.1 Setup

**Base Models and Benchmark.** In line with DD1 [21], we choose VAR [37] and LlamaGen [35] as the base AR models due to their popularity and strong generation quality. Moreover, these two models differ significantly across several key aspects, which makes them ideal testbeds for evaluating the generality of DD2: **(1) Tokenizer training**: VAR's codebook is trained to support multi-resolution image tokens, while LlamaGen's tokens are derived solely from the original resolution space; **(2) Token ordering**: VAR constructs the full sequence by concatenating sub-sequences across different resolutions, whereas LlamaGen follows a traditional raster-scan order; **(3) Generation steps**: VAR has 10 sampling steps, while LlamaGen requires 256 steps. These differences allows us to evaluate how DD2 performs across a wide range of AR setups. We choose the popular and standard ImageNet-256 dataset as the benchmark.

**Generation.** We use the one-step sample quality as our main results in Tab. 1. Additionally, following DD1, we involve the teacher AR model in sampling for smoother trade-off of quality and steps. Results are listed in Tab. 2.

**Baselines.** Since DD1 [21] is the only method that enables few-step sampling for image AR models, we take it as our main baseline. We also report the results of several weak baselines in the DD1 paper: **(1)** directly skip last several steps, and **(2)** predicting the distribution of all tokens in one step, which is the extreme case of set-of-token prediction method. Details of baseline can be found in App. E.5.

## 5.2 Results of One-step Generation

We demonstrate the main results of DD2 in Tab. 1. Since the model parameter sizes and inference latency of DD2 and DD1 are similar, we compare them under the same number of sampling steps. The key takeaways are:

**The performance gap between DD2 and the teacher AR model is minimal.** For VAR models across all model sizes, compressing the teacher model to 1 step and achieving up to $8.1\times$ speedup with an mere FID increase of less than 2.5. For LlamaGen models, DD2 achieves $238\times$ speed-up in a FID increase of only 4.48. Such a performance drop is acceptable.

**DD2 outperforms the strongest baseline DD1 significantly**. For VAR models across all model sizes, DD2 decreases the performance gap between the teacher AR model and one-step model by up to 67% compared to DD1. DD2 even outperforms the 2 step sampling results of DD1 by a large margin for all VAR models. For LlamaGen model, DD2 also achieves a 2.76 better FID than DD1. All weak baselines fail to generate in one-step. These results show the effectiveness of DD2.

## 5.3 Results of Multi-step Sampling

It is better to offer a smoother trade-off curve between quality and step. To achieve this, we use the teacher model to refine the last several AR positions of the generated content. The detailed algorithm is shown at Alg. 4. Results are reported in Tab. 2. For DD1 baseline, we use its default multistep sampling schedule and control the number of sampling steps to ensure a fair comparison. The sample quality increases consistently with more sampling steps, offering more choices for the users.

## 5.4 Training Efficiency

In addition to its superior performance over DD1, DD2 offers another significant advantage: much faster convergence. As shown in Tab. 3, DD2 requires substantially much fewer GPU hours to train, achieving up to a $12.3\times$ training speedup while having better performance than DD1. Detailed analysis of DD2's training cost can be found at App. B.4.

Table 1: Generative performance on class-conditional ImageNet-256. "#Step" indicates the number of model inference to generate one image. "Time" is the wall-time of generating one image in the steady state. Results with [†] are taken from [21], while * denotes results obtained with more training.

| Type | Model | FID↓ | IS↑ | Pre↑ | Rec↑ | #Para | #Step | Time |
|---|---|---|---|---|---|---|---|---|
| GAN[†] | StyleGan-XL [30] | 2.30 | 265.1 | 0.78 | 0.53 | 166M | 1 | 0.3 |
| Diff.[†] | ADM [5] | 10.94 | 101.0 | 0.69 | 0.63 | 554M | 250 | 168 |
| Diff.[†] | LDM-4-G [28] | 3.60 | 247.7 | – | – | 400M | 250 | – |
| Diff.[†] | DiT-L/2 [25] | 5.02 | 167.2 | 0.75 | 0.57 | 458M | 250 | 31 |
| Mask.[†] | MaskGIT [2] | 6.18 | 182.1 | 0.80 | 0.51 | 227M | 8 | 0.5 |
| AR[†] | VQGAN [6] | 15.78 | 74.3 | – | – | 1.4B | 256 | 24 |
| AR[†] | ViTVQ [45] | 4.17 | 175.1 | – | – | 1.7B | 1024 | >24 |
| AR[†] | RQTran. [15] | 7.55 | 134.0 | – | – | 3.8B | 68 | 21 |
| AR | VAR-d16 [37] | 4.15 | 278.7 | 0.85 | 0.41 | 310M | 10 | 0.133 |
| AR | VAR-d20 [37] | 3.40 | 305.1 | 0.84 | 0.47 | 600M | 10 | 0.184 |
| AR | VAR-d24 [37] | 2.86 | 312.9 | 0.82 | 0.51 | 1.03B | 10 | 0.251 |
| AR | LlamaGen-L [35] | 4.11 | 283.5 | 0.85 | 0.48 | 343M | 256 | 5.01 |
| Weak Baseline[†] | VAR-*skip-2* | 40.09 | 56.8 | 0.46 | 0.50 | 310M | 8 | 0.098 |
| Weak Baseline[†] | VAR-*onestep** | 157.5 | – | – | – | – | 1 | – |
| Weak Baseline[†] | LlamaGen-*skip-156* | 80.72 | 12.13 | 0.17 | 0.20 | 343M | 100 | 1.95 |
| Weak Baseline[†] | LlamaGen-*onestep** | 220.2 | – | – | – | – | 1 | – |
| DD1 | VAR-d16 | 9.94 | 193.6 | 0.80 | 0.37 | 327M | 1 | 0.021 |
| DD1 | VAR-d16 | 7.82 | 197.0 | 0.80 | 0.41 | 327M | 2 | 0.036 |
| DD1 | VAR-d20 | 9.55 | 197.2 | 0.78 | 0.38 | 635M | 1 | 0.027 |
| DD1 | VAR-d20 | 7.33 | 204.5 | 0.82 | 0.40 | 635M | 2 | 0.047 |
| DD1 | VAR-d24 | 8.92 | 202.8 | 0.78 | 0.39 | 1.09B | 1 | 0.034 |
| DD1 | VAR-d24 | 6.95 | 222.5 | 0.83 | 0.43 | 1.09B | 2 | 0.059 |
| DD1 | LlamaGen-L | 11.35 | 193.6 | 0.81 | 0.30 | 326M | 1 | 0.023 |
| DD1 | LlamaGen-L | 7.58 | 237.5 | 0.84 | 0.37 | 326M | 2 | 0.043 |
| DD2 (ours) | VAR-d16 | 6.21 | 213.0 | 0.84 | 0.39 | 329M | 1 | 0.019 (7.0×) |
| DD2 (ours) | VAR-d20 | 5.43 | 233.7 | 0.85 | 0.41 | 619M | 1 | 0.023 (8.0×) |
| DD2 (ours) | VAR-d24 | 5.06 | 254.7 | 0.85 | 0.39 | 1.04B | 1 | 0.031 (8.1×) |
| DD2 (ours) | VAR-d24* | 4.91 | 282.2 | 0.87 | 0.39 | 1.04B | 1 | 0.031 (8.1×) |
| DD2 (ours) | LlamaGen-L | 8.59 | 229.1 | 0.77 | 0.32 | 335M | 1 | 0.021 (238×) |
| DD2 (ours) | LlamaGen-L* | 7.58 | 238.7 | 0.77 | 0.34 | 335M | 1 | 0.021 (238×) |

Table 2: Generation quality of involving the pre-trained AR model when sampling. The notation *pre-trained-n-m* means that the pre-trained AR model is used to re-generate the $n+1$-th to $m$-th tokens in the sequence generated in the first step by the few-step generator.

| Type | Model | FID↓ | IS↑ | Pre↑ | Rec↑ | #Para | #Step |
|---|---|---|---|---|---|---|---|
| AR | VAR-d16 [37] | 4.19 | 230.2 | 0.84 | 0.48 | 310M | 10 |
| DD1 | VAR-d16-*pre-trained-4-5* | 6.54 | 210.8 | 0.83 | 0.42 | 327M | 3 |
| DD1 | VAR-d16-*pre-trained-3-5* | 5.47 | 230.5 | 0.84 | 0.43 | 327M | 4 |
| DD1 | VAR-d16-*pre-trained-0-5* | 5.03 | 242.8 | 0.84 | 0.45 | 327M | 6 |
| DD2 (ours) | VAR-d16-*pre-trained-8-10* | 5.24 | 238.9 | 0.85 | 0.40 | 329M | 3 |
| DD2 (ours) | VAR-d16-*pre-trained-7-10* | 4.88 | 248.7 | 0.86 | 0.41 | 329M | 4 |
| DD2 (ours) | VAR-d16-*pre-trained-5-10* | 4.47 | 277.8 | 0.87 | 0.42 | 329M | 6 |

## 5.5 Ablation Study: the Importance of Initialization

As discussed in Sec. 3.3, initialization is dispensable for DD2. We provide results on LlamaGen-L and VAR-d24 models to verify this, by only initializing one of the generator and guidance network with the tuned AR-diffusion model, while the other uses the backbone from the teacher AR model and a randomly initialized output head. Results are shown in Tab. 4. We find that missing proper initialization for either of them can lead to significant performance degradation or even collapse, highlighting the importance of good initialization for both components.

## 6 Discussions

### 6.1 Distinction with diffusion score distillation

In this section, we discuss the differences between score distillation for diffusion models and DD2.

**Fundamentally Different Task.** Traditional score distillation methods aim to reduce the number of sampling steps for *diffusion models*. In contrast, our work focuses on one-step sampling for pre-trained *AR models*, which is a fundamentally different generative paradigm from diffusion models. Despite the competitive or even superior performance of AR models compared to diffusion models, one-step sampling for AR models is under explored, which highlights the contribution of DD2.

Table 3: Training cost of DD2 and speed-up compared with DD1. All experiments are done on 8 NVIDIA A800 GPUs.

| Method | Model | Param | Cost (8×GPU h) | Speed-up |
|---|---|---|---|---|
| DD1 | VAR-d16 | 327M | 296.9 | 1× |
| DD1 | VAR-d20 | 635M | 484.4 | 1× |
| DD1 | VAR-d24 | 1.09B | 604.2 | 1× |
| DD1 | LlamaGen-L | 326M | 647.7 | 1× |
| DD2 (ours) | VAR-d16 | 329M | 115.5 | 2.6× |
| DD2 (ours) | VAR-d20 | 619M | 174.4 | 2.8× |
| DD2 (ours) | VAR-d24 | 1.04B | 96.1 | 6.3× |
| DD2 (ours) | LlamaGen-L | 335M | 52.6 | 12.3× |

Table 4: Impact of Initialization.

| Gui-Init | Gen-Init | Model | Param | FID-5k |
|---|---|---|---|---|
| ✓ | ✓ | LlamaGen-L | 335M | **14.77** |
| ✓ | × | LlamaGen-L | 335M | 16.08 |
| × | ✓ | LlamaGen-L | 335M | 21.76 |
| ✓ | ✓ | VAR-d24 | 1.04B | **11.53** |
| ✓ | × | VAR-d24 | 1.04B | Collapse(>200) |
| × | ✓ | VAR-d24 | 1.04B | Collapse(>200) |

Table 5: Perceptual path length of DD2 and DD1.

| | DD1 | DD2 |
|---|---|---|
| PPL↓ | 18437.6 | 7231.9 |

**Technical Adaptations.** Directly applying standard score distillation to AR models is not feasible. We have made multiple technical innovations to tackle this problem: (1) we train the guidance network to learn the *conditional score* of the generator instead of the score, (2) we replace the classification layer in AR models with MLP head to ensure continuous output, and (3) we propose to adapting the pre-trained AR model into an AR-diffusion model as an initialization. We further replace the standard AR-diffusion loss with our GTS loss for better convergence of this process.

Our strong results offer a new perspective on training one-step generative models. Currently, the dominant strategy for training such models focuses on diffusion-based frameworks. In contrast, our method demonstrates that distilling an AR model is also a highly competitive approach, as our results surpass many representative diffusion distillation techniques shown in Tab. 6.

### 6.2 Benefits of Eliminating Pre-defined Mapping in DD1

Compared to DD1, a key feature of DD2 is that it does not rely on any pre-defined mapping, which brings several potential benefits: **(1) More efficient utilization of model knowledge.** In DD1, the pre-defined mapping provides only a single end-to-end signal, whereas DD2 explicitly trains the model at every token position, offering a more fine-grained supervisory signal. **(2) Reduced accumulation of errors.** In DD1, if the model fails to correctly learn the noise-to-data mapping at a certain position, this error propagates to subsequent positions because their conditions depend on earlier predictions. In contrast, in DD2, the teacher model provides ground-truth distributions for each token position based on the generator's current condition. Since the teacher model possesses strong generalization ability, the impact of imperfect conditions is greatly mitigated. (3) **Smoother latent representations.** More generally, training generative models without pre-defined mappings allows them to automatically discover smoother latent representations of the target data distribution, which benefits learning because smoother representations are easier to optimize. To quantify this property, we measure the Perceptual Path Length (PPL) metric [12] for both DD2 and DD1, where a lower value indicates smoother interpolation in the latent space. As shown in Tab. 5, DD2 achieves significantly smoother latent interpolation than DD1.

## 7 Future Works and Limitations

**Compatibility with Image AR Models without VQ.** In addition to the commonly used discrete-space AR models based on VQ hidden space, continuous-space AR models [17] has recently gained increasing popularity. These models generate each token through a diffusion process. Our method is naturally compatible with such models as well, since they directly provide the conditional score. We leave the application of our approach to this class of models as future work.

**Scaling to Larger Tasks.** Image AR models have also been used in larger-scale tasks, such as text-to-image task [9, 47]. Extending our method to these models offers practical impact.

**Performance Gap to the Teacher Model.** Although DD2 achieves significant speedup, the distilled models still exhibit a certain performance gap compared to the original AR models. Addressing this performance drop to make one-step AR models match or even surpass the quality of pretrained AR models remains an important and promising direction for future research.

## Acknowledgement

This work was supported by National Natural Science Foundation of China (62506197, No. 62325405, 62104128, U19B2019, U21B2031, 61832007, 62204164, 92364201), Tsinghua EE Xilinx AI Research Fund, and Beijing National Research Center for Information Science and Technology (BNRist). We would like to thank all anonymous reviewers for their suggestions. We also thank all the support from Infinigence-AI.

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

# A  Discussion of Prop. 1

In this section, we provide the proof of Prop. 1 using a simple induction.

*Proof.* Assuming the neural network has sufficiently large capacity, minimizing $\mathcal{L}_{CSD}$ is then equivalent to minimizing each individual term $\mathcal{L}_{CSDi} = d(s_\Phi(q_i^{t_i}, t_i|sg(q_{<i})), s_{fake}(q_i^{t_i}, t_i|sg(q_{<i})))$ for any $i$.

**Base case** ($i = 1$): At the first position $i = 1$, the corresponding term $\mathcal{L}_{CSD1} = d(s_\Phi(q_1^{t_1}, t_1), s_{fake}(q_1^{t_1}, t_1))$ degrades to a traditional score distillation loss, so that minimizing this term guarantees $p_\theta(q_1) = p_\Phi(q_1)$.

**Inductive Hypothesis** ($i = k - 1$): Assume that for token position $i = k$, the generator correctly models the distribution of all tokens before this position: $p_\theta(q_{<k}) = p_\Phi(q_{<k})$.

**Inductive Step** ($i = k$) Minimizing $\mathcal{L}_{CSDk} = d(s_\Phi(q_k^{t_k}, t_k|sg(q_{<k})), s_{fake}(q_k^{t_k}, t_k|sg(q_{<k})))$ guarantees $p_\theta(q_k|q_{<k}) = p_\Phi(q_k|q_{<k})$, so that $p_\theta(q_{<k+1}) = p_\theta(q_k|q_{<k})p_\theta(q_{<k}) = p_\Phi(q_k|q_{<k})p_\Phi(q_{<k}) = p_\Phi(q_{<k+1})$ holds

**Conclusion**: By mathematical induction, minimizing $\mathcal{L}_{CSD}$ guarantees $p_\theta(q_1, \ldots, q_n) = p_\Phi(q_1, \ldots, q_n)$.

$\square$

# B  More Experimental Details

In this section, we provide some additional experimental settings and results.

## B.1  Comparison with Diffusion Distillation Methods

We compare DD2 with several commonly used diffusion distillation methods to show our effectiveness. Results are shown in Tab. 6.

Table 6: Comparison between DD2 and diffusion distillation methods. Results are taken from the paper of Shortcut model [7].

|  | DD2-VAR-d16 | DD2-VAR-d20 | DD2-VAR-d24 | PD[29] | CD[33] | CT[33] | Reflow[22] | Shortcut[7] |
|---|---|---|---|---|---|---|---|---|
| FID↓ | 6.21 | 5.43 | 4.91 | 35.6 | 136.5 | 69.7 | 44.8 | 10.6 |

## B.2  Training Curve of the Original DMD Method

We demonstrate our reproduced FID-iteration curve of the original DMD method with different initialization setting in Fig. 6. As discussed in Sec. 3.3, an inappropriate score distillation initialization strategy can lead to slower convergence and bad training stability, which motivates us to use the AR-diffusion model for initialization.

## B.3  Performance of the AR-diffusion Model for Initialization

Table 7: Performance of AR-diffusion models. FIDs are evaluated wit 5k generated images.

| Loss | Model | Param | FID-5k | Training Iter |
|---|---|---|---|---|
| GTS | VAR-d16 | 329M | 11.41 | 330k |
| GTS | VAR-d20 | 619M | 11.41 | 330k |
| GTS | VAR-d24 | 1.04B | 11.24 | 230k |
| GTS | LlamaGen-L | 335M | 15.66 | 100k |
| diffusion | VAR-d16 | 329M | 17.98 | 500k |

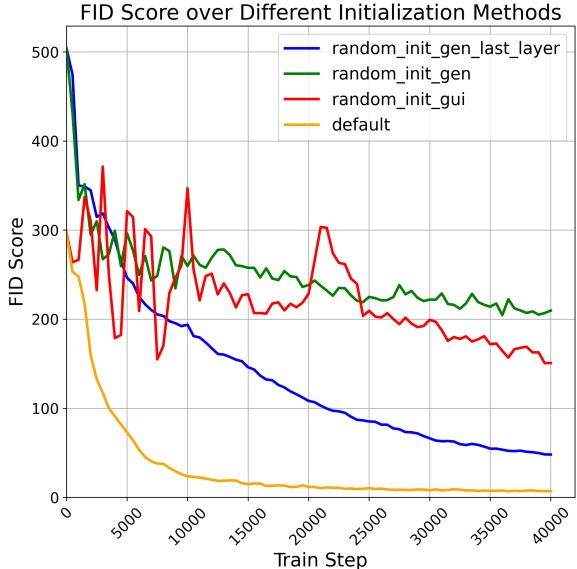

Figure 6: Training results for different initialization strategies. *default* indicates that both the generator and the guidance components are initialized using the teacher model [43], *random_init_gen_last_layer* refers to the setting where only the last layer of the generator is randomly initialized, while the rest of the generator and the guidance are initialized from the teacher model. *random_init_gen* means the entire generator is randomly initialized, whereas the guidance is initialized from the teacher model. *random_init_gui* denotes the case where only the guidance is randomly initialized, with the generator initialized from the teacher model.

Table 8: Wall time (hour) of each training stage under different settings. All time costs are profiled on 8 NVIDIA A800 GPUs.

| Model | VAR-d16 | VAR-d20 | VAR-d24 | LlamaGen-L |
|---|---|---|---|---|
| Continuous Adaptation | - | - | - | 3.3 |
| AR-diffusion Tuning (Head Only) | 60.1 | 72.6 | 59.9 | - |
| AR-diffusion Tuning (Full Model) | 8.0 | 9.1 | 17.2 | 8.6 |
| Main Training | 42.6 | 88.0 | 19.0 | 40.7 |
| Performance Alignment | 4.8 | 4.7 | - | - |
| Overall | 115.5 | 174.4 | 96.1 | 52.6 |

We report the performance of the tuned AR-diffusion Model in Tab. 7, which serves as the initialization of both generator and guidance network. We use a 10-step Euler solver for the sampling process of every token. To verify the effectiveness of the proposed GTS loss Eq. (6), we also report the results of using traditional diffusion loss where the target is a Monte Carol of the ground truth score. The key takeaways are: **(1)** the tuned AR-diffusion model demonstrates strong sample quality, making it a suitable choice for initialization, and **(2)** performance degrades significantly if we use traditional diffusion loss, highlighting the effectiveness and necessity of our GTS loss.

## B.4 Training Details and Cost

The training mainly cost consists of two parts: AR-diffusion tuning and the main training process with CSD loss.

In the first stage, we use the same setting as in evaluation to generate the training data. Specifically, for all VAR models, we apply *top_k* as 900 and *classifier-free-guidance* scale as 2.0; for LlamaGen model we use *top_k* as 8000 and *classifier-free-guidance* scale as 2.0.

Table 9: Converged performance before/after using performance alignment operation.

| Type | Model | FID↓ | IS↑ | Pre↑ | Rec↑ | #Para | #Step |
|---|---|---|---|---|---|---|---|
| DD2 (Before) | VAR-d16 | 8.35 | 176.7 | 0.80 | 0.42 | 329M | 1 |
| DD2 (Before) | VAR-d20 | 6.57 | 201.4 | 0.81 | 0.43 | 618M | 1 |
| DD2 (After) | VAR-d16 | 6.21 | 213.0 | 0.84 | 0.39 | 329M | 1 |
| DD2 (After) | VAR-d20 | **5.43** | **233.7** | **0.85** | 0.41 | 618M | 1 |

For VAR models, in the first stage, we first fine-tune only the output head, and subsequently train the entire model. For the second part, we apply performance alignment to d16 and d20 models, which takes additional cost of training the guidance network. We list the cost of each part in Tab. 8.

For LlamaGen model, there is an additional stage where we fine-tune the teacher model to support continuous input. For AR-diffusion tuning, we directly train the entire model. We don't apply performance alignment. Detailed results are listed in Tab. 8.

### B.5 The Effectiveness of Performance Alignment

We provide the convergence performance of the d16 and d20 models before and after performing the performance alignment procedure in Tab. 9, demonstrating the effectiveness of this operation.

## C  Implementation Techniques

In this section, we introduce several techniques we use in our pipeline.

### C.1  Computing Score Function with Probability Vector

In this section, we derive Eq. (1) starting from the preliminaries of flow matching.

Flow matching [20, 22] defines an invertible transformation with ordinary differential equation (ODE) $dx = V(x_t, t)dt$ between two distributions $\pi_0(x)$ and $\pi_1(x)$. The velocity function under linear noise schedule $x_t = (1 - t)x_0 + tx_1$ can be given as:

$$V(x_t, t) = \mathbb{E}_{x_0, x_1 \sim \pi_{0,1}(x_0, x_1)}(x_1 - x_0 | (1 - t)x_0 + tx_1 = x_t), \qquad (7)$$

where $\pi_{0,1}(x_0, x_1)$ is any joint distribution that satisfies temporal boundary conditions at both ends: $\int \pi_{0,1}(x_0, x_1)dx_0 = \pi_1(x_1)$ and $\int \pi_{0,1}(x_0, x_1)dx_1 = \pi_0(x_0)$.

Since the source distribution $\pi_1(x)$ is a Gaussian distribution here, the relationship between the score function $s(x_t, t)$ and the velocity $V(x_t, t)$ is as follows:

$$v(x_t, t) = -\frac{ts(x_t, t) + x_t}{1 - t} \qquad (8)$$

And the score function can be given as:

$$s(x_t, t) = \mathbb{E}_{x_0, x_1 \sim \pi_{0,1}(x_0, x_1)}(-\frac{x_1}{t} | (1 - t)x_0 + tx_1 = x_t) \qquad (9)$$

In our problem, the target distribution is a weighted sum of Dirac functions: $\pi_0(x) = \sum_{j=1}^{V} p_j \delta(x - c_j)$, and is independent of the source distribution, resulting in only a finite number of possibilities. Therefore, we only need compute the product of the source distribution probability and the target distribution probability for each possible case, and then use this as the weight function to compute the expectation of $-\frac{x_1}{t}$. With the above explanations, we can easily arrive at Eq. (1).

### C.2  Multiple Noisy Samples for Fake Conditional Score Learning

As discussed in Sec. 3.2.2, the output of the guidance network $s_\psi(q_i^{t_i}, i | q_{<i})$ is computed in two stages. First, a transformer backbone process the input sequence $(q_1, \ldots, q_n)$ and outputs a feature sequence $(f_1, \ldots, f_n)$, where each $f_i$ is causally conditioned on $q_{<i}$. Then, a lightweight MLP takes

the noisy token $q_i^{t_i}$, timestep $t_i$ and the feature $f_i$ as input, then outputs the estimated conditional score $s_\psi(q_i^{t_i}, t_i | q_{<i})$.

Every conditioning $f_i$ defines a continuous distribution over noisy inputs $(x_t, t)$, and the model must learn to predict score function across the entire space and all timesteps. To ensure sufficient training and improve generalization, we draw inspiration from the MAR implementation (see `https://github.com/LTH14/mar` for more details) and apply a multi-sample training strategy. Specifically, for a generated sequence $(q_1, \ldots, q_n)$, we sample multiple noise sequences $(\epsilon_1, \ldots, \epsilon_n)_{1,\ldots,m}$ to create multiple noisy versions of the generated sequence. For each noisy sequence, we apply Eq. (5) as the loss function and take the average across all $m$ samples. The resulting training objective is:

$$\mathcal{L}_{FCS} = \mathbb{E}_{(q_1,\ldots,q_n) \sim p_\theta} \sum_{j=1}^{m} \sum_{i=1}^{n} \|s_\psi(q_i^{t_{i,j}}, t_{i,j} | q_{<i}) - \nabla_{q_i^{t_{i,j}}} \log p(q_i^{t_{i,j}} | q_i)\|^2, \tag{10}$$

where $t_{i,j}$ is the $j$-th sample at the $i$-th token position, $q_i^{t_{i,j}} = (1 - t_{i,j})q_i + t_{i,j}\epsilon_{i,j}$, with $\epsilon \sim \mathcal{N}(0; \mathbf{I})$ denotes the $j$-th noise sample at the $i$-th token position.

### C.3 Performance Alignment for both Generator and Guidance network

We find there are two issues during training: **(1)** there is a discrepancy between the guidance network and the generator's score, and **(2)** unstable training dynamics. Specifically, we observe that the samples generated by the conditional guidance network via AR-diffusion tend to under-perform those generated by the generator, indicating a training gap of the guidance network. Additionally, we notice that the generator's FID fluctuates significantly during training. However, applying the Exponential Moving Average (EMA) technique to the generator leads to a much more stable performance curve. These findings motivate us to introduce a **performance alignment** procedure for both generator and guidance network after an initial phase of training. Specifically, **(1)** for the generator, we replace the regular model weights directly with EMA weights, then **(2)** we fix the generator and only train the conditional guidance network for a certain period to adapt it to the new generator distribution. Once this alignment process is complete, we resume the standard training process. We empirically found that this technique is particularly helpful for training VAR-d16 and VAR-d20 models, significantly improving performance even after the models have already converged. Results are shown at Tab. 9.

### C.4 Larger Update Frequency for the Guidance Network

Training the guidance network is crucial, as it is responsible for producing accurate conditional scores of the generator distribution. However, this task is challenging because the generator distribution is also evolving during training. To address this issue, we adopt a higher update frequency for the guidance network, following the strategy used in DMD2 [42]. Specifically, in each training iteration, we update the guidance network $K$ times with $(K > 1)$ while updating the generator only once. The specific values of $K$ used for different models are provided in App. E.

### C.5 Details of Model Architectures

We use the same architecture for both generator and guidance network. Inspired by MAR [17], our model architecture consists of a transformer backbone and a lightweight MLP head. As discussed in Sec. 3, for the guidance network, the transformer backbone takes the token sequence as input and output a causal feature sequence, while the MLP head takes the feature, noisy token and timestep as input and output the predicted conditional score. For the generator, the transformer backbone takes the shifted noise sequence as input, while the MLP head takes the noise sequence and the feature sequence as input and give the final generated sequence. We demonstrate the model architectures in Fig. 7.

## D  Algorithms of Multi-step Sampling Method

In this section, we present the pseudo algorithm of the multi-step sampling method in Sec. 3. Suppose we have a sequence $X = (q_1, \ldots, q_n)$. We denote the indexing operations $X[t] = q_t$ and $X[:t] = (q_1, \ldots, q_{t-1})$. The pseudo algorithm is presented in Alg. 4, with results reported in Tab. 2.

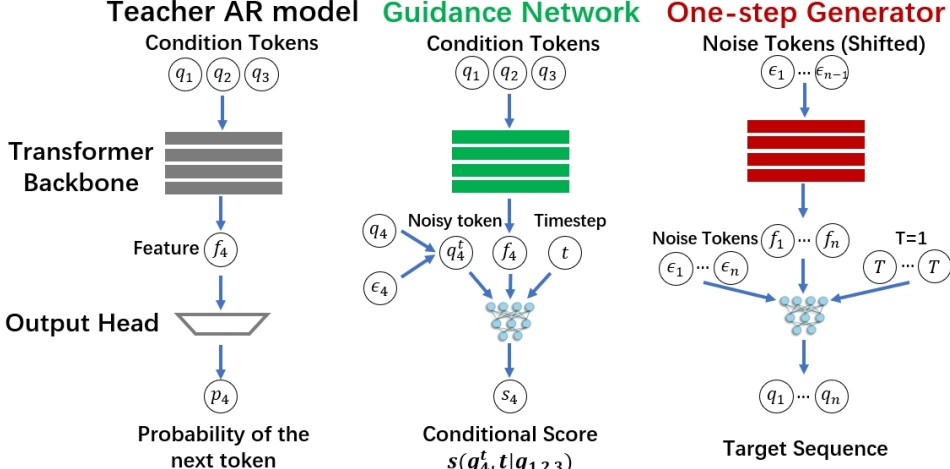

Figure 7: Demonstration of the model architectures and the corresponding inputs/outputs.

Table 10: Hyperparameters used for AR-diffusion model tuning in the initialization phase. Actual BS/iter refers to the actual batch size used in each training iteration. The training iterations for VAR are reported in the format of 'only head + full model' to reflect the two-phase training procedure.

| Hyperparameter | VAR-d16 | VAR-d20 | VAR-d24 | LlamaGen-L |
|---|---|---|---|---|
| Learning Rate | 2e-4 | 2e-4 | 2e-4 | 1e-4 |
| Batch Size | 512 | 512 | 512 | 512 |
| Grad Accumulation | 4 | 8 | 16 | 4 |
| Actual BS/iter | 128 | 64 | 32 | 128 |
| $Adam\ \beta_0$ | 0.9 | 0.9 | 0.9 | 0.9 |
| $Adam\ \beta_1$ | 0.95 | 0.95 | 0.95 | 0.95 |
| Training Iterations | 300k+30k | 300k+30k | 200k+30k | 100k |

---

**Algorithm 4:** Sampling with the teacher AR model

---

**Require:** : The distilled one-step model $\theta$, the pre-trained AR model $\Phi$, total sampling steps $k > 1$.
**Sampling Process**
1: $X = (q_1, \ldots, q_n) \leftarrow$ one-step sampling by $\theta$
2: **for** $t$ in $\{n - k + 2, \ldots, n\}$ **do**
3:     Sample $q'_t \sim p_\Phi(\cdot | X[: t])$
4:     $X[t] \leftarrow q'_t$
5: **end for**
6: **return** $X$

---

# E  Detailed Experimental Settings

In this section, we present the settings of our training process.

## E.1  Model Parameterization

In this work, we parameterize the network as velocity prediction network, due to its widely verified good properties. Specifically, we use $v(x_t, t) = -\frac{\sigma_t s(x_t, t) - x_t}{\alpha_t}$ to convert between velocity and score. We use the velocity function to train the AR-diffusion model and the guidance network. For the generator, we use $x_\theta = \epsilon - v_\theta$ as its final output, where $\epsilon$ is the input noise and $v_\theta$ is the model output.

Table 11: Hyperparameters used for DD2. Actual BS/iter refers to the actual batch size used in each training iteration. "Gen" and "Gui" stand for the generator and guidance network, respectively. The training iterations of the generator for VAR-d16 and VAR-d20 are reported in the format of "before performance alignment + after performance alignment", while the training iterations of the guidance network for VAR-d16 and VAR-d20 are reported in the format of "before performance alignment + alignment + after performance alignment".

| Hyperparameter | VAR-d16 | | VAR-d20 | | VAR-d24 | | LlamaGen-L | |
| --- | --- | --- | --- | --- | --- | --- | --- | --- |
| | Gui $\psi$ | Gen $\theta$ | Gui $\psi$ | Gen $\theta$ | Gui $\psi$ | Gen $\theta$ | Gui $\psi$ | Gen $\theta$ |
| Start Learning rate | 1e-6 | 1e-6 | 1e-6 | 1e-6 | 1e-6 | 1e-6 | 1e-6 | 1e-6 |
| End Learning rate | 2e-4 | 2e-4 | 2e-4 | 2e-4 | 4e-4 | 5e-5 | 2e-4 | 1e-4 |
| Batch size | 512 | 512 | 512 | 512 | 1024 | 1024 | 1024 | 1024 |
| Grad Accumulation | 4 | 4 | 8 | 8 | 32 | 32 | 8 | 8 |
| Actual BS/Iter | 128 | 128 | 64 | 64 | 32 | 32 | 128 | 128 |
| $Adam\ \beta_0$ | 0.9 | 0.9 | 0.9 | 0.9 | 0.9 | 0.9 | 0.9 | 0.9 |
| $Adam\ \beta_1$ | 0.95 | 0.95 | 0.95 | 0.95 | 0.95 | 0.95 | 0.95 | 0.95 |
| Training iterations | 40k+8k+15k | 8k+7.5k | 80k+8k+36k | 16k+18k | 30k | 15k | 60k | 12k |
| Guidance Update Freq | 5(Before Align), 2(After Align) | | 5(Before Align), 2(After Align) | | 2 | | 5 | |

## E.2 Stage 1: AR-diffusion Tuning

For VAR models, we first freeze the transformer backbone and tune the output MLP for certain iterations. Then we remove the constraints and tune all parameters. For LlamaGen model, we directly tune all parameters from the start. Settings are listed in Tab. 10.

## E.3 Stage 2: Training with CSD Loss

**Calculation of the Real Conditional Score** We follow the default sampling settings of original AR models for probability vector calculation. For VAR models, we set classifier-free guidance scale to 2.0, top-k to 900 and top-p to 0.95. For LlamaGen model, we set classifier-free guidance scale to 2.0, top-k to 900 and top-p to 1.0.

**Optimization Settings** We list the optimization settings in Tab. 11. For the learning rate of the generator, we apply a linear warm up strategy from the start learning rate to the end learning rate in 40K guidance network training iterations for VAR-d16, VAR-d20 and LlamaGen-L models. For VAR-d24, we set the warm up length as 20K.

**EMA** We find that EMA is very important for the stability of training. Since the model performs badly at the beginning of the training process, we use a progressive EMA rate. Specifically, we use a small EMA rate in the early stage of training. Then we use a dynamic EMA rate $min(0.9999, (iter + 1)/(iter + 10))$, where $iter$ is the training iteration. This progressive EMA schedule ensures both training stability and fast convergence.

## E.4 Continuous Input Adaptation for LlamaGen Models

For VAR teacher models, we directly conduct our workflow since they naturally support continuous embeddings as input. However, for LlamaGen teacher model, we need to modify the model's input head to accept continuous latent embeddings instead of discrete token indices. This allows the model to handle the continuous outputs from the generator. Specifically, we replace the original $nn.embedding$ layer $emb$ with a MLP $mlp$. We first train this MLP with loss $\sum_{i=1}^{V} \|mlp(c_i) - emb(i)\|^2$, where $i$ is the token and $V$ is the total number of tokens in the codebook. This process is fast, but incurs performance loss. Then we fine-tune the whole model with standard AR loss implemented by LlamaGen for 200K iterations, to align its performance with the original model. Finally, we obtained an AR LlamaGen model that incurs no performance loss and supports continuous embeddings as input, which we use as the teacher model.

## E.5 Baselines

**Set Prediction Method.** Set prediction is a commonly used technique for reducing the sampling steps of Image AR models [2, 37, 17]. However, it is fundamentally incapable of reducing the sampling process to a single step. As pointed out in the DD1 paper [21], when training a model under the one-step sampling setting with this approach, the optimal solution is equivalent to *independently sampling* each token according to the overall token frequency at this position in the dataset. Therefore,

we can directly evaluate its one-step generation performance without actually training a model. This method completely ignores the dependencies between tokens, which leads to the failure case reported in the *onestep\** rows of Tab. 1.

**Pre-trained AR Models.** For VAR models, we set *top_k*, *top_p* and *classifier-free-guidance scale* as 900, 0.95 and 1.5, respectively. For LlamaGen models, we use 8000 as *top_k*, 1.0 as *top_p*, and 2.0 as *classifier-free-guidance scale*

# F  Visualizations

We show some generated examples in Figs. 8 to 11.

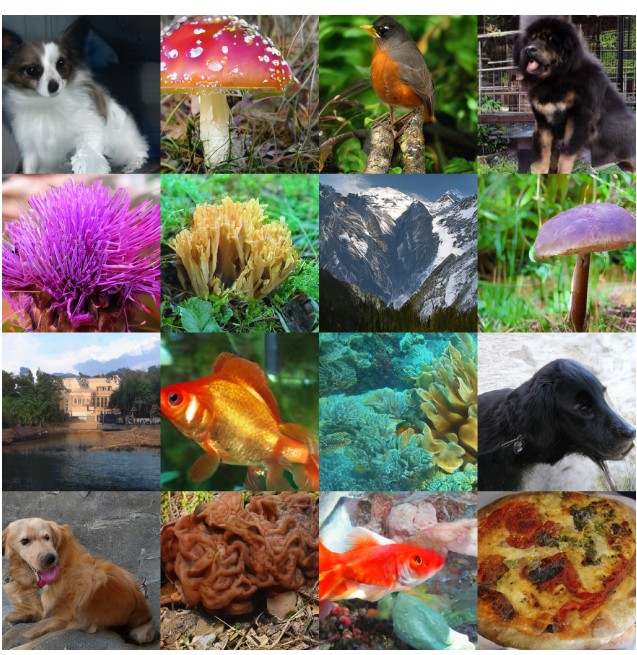

Figure 8: Generated by DD2-VAR-d16 model.

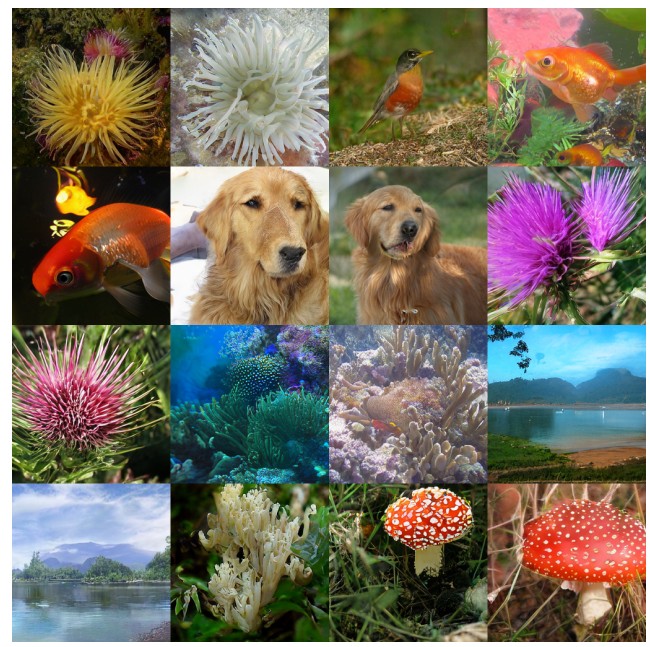

Figure 9: Generated by DD2-VAR-d20 model.

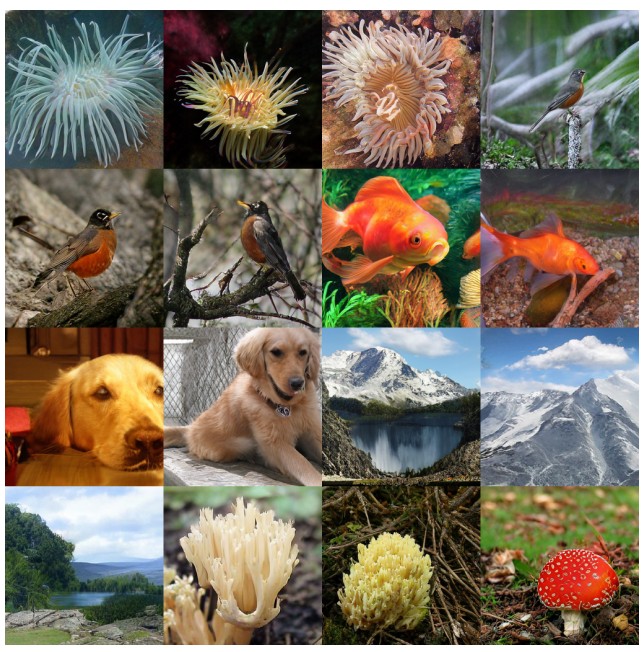

Figure 10: Generated by DD2-VAR-d24 model.

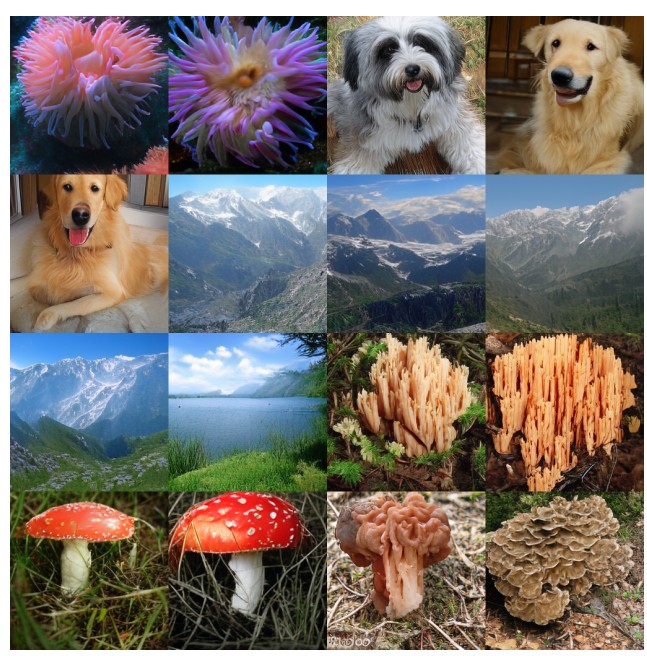

Figure 11: Generated by DD2-LlamaGen-L model.

