# OpenReview forum: "Distilled Decoding 2: One-step Sampling of Image Auto-regressive Models with Conditional Score Distillation"
_NeurIPS.cc/2025/Conference — NeurIPS 2025 poster_

### Official Review · Reviewer_qqvy · 2025-06-30

**Clarity:** 3
**Significance:** 3
**Originality:** 4
**Rating:** 4
**Confidence:** 4

**Summary:**

This paper introduces Distilled Decoding 2 (DD2), a novel method for high-quality, one-step sampling of powerful but slow Image Auto-regressive (AR) models. DD2 re-interprets the teacher AR model as a conditional score provider and uses a novel Conditional Score Distillation (CSD) loss to train a one-step generator. This approach avoids the rigid mappings of prior work and significantly closes the performance gap to the original multi-step model, outperforming previous methods while also being much more efficient to train.

**Questions:**

See Weakness

**Ethical Concerns:**

["NO or VERY MINOR ethics concerns only"]

**Final Justification:**

The authors have addressed my concerns, and I maintain my stance of accepting the paper. The authors should incorporate the revised content into the final version.

**Limitations:**

Yes. The authors have a dedicated "Future Works and Limitations" section (Section 6) where they transparently discuss the remaining performance gap to the teacher model and identify promising directions for future work, such as applying the method to other types of AR models and larger tasks. The discussion is constructive and appropriate.

**Quality:**

3

**Strengths And Weaknesses:**

Strengths

1.	The paper tackles a highly significant problem in generative modeling. Image AR models produce state-of-the-art results but are hindered by extremely slow, sequential inference. Enabling high-fidelity one-step generation makes these powerful models far more practical for real-world applications. The substantial improvement over the previous state-of-the-art (DD1) marks a major step forward in this domain.

2.	The experimental results are comprehensive and convincing. The method is evaluated on a standard, large-scale benchmark (ImageNet-256) using multiple strong and diverse AR base models (VAR and LlamaGen). The comparisons against DD1 and other baselines are fair and clearly demonstrate the superiority of DD2 in terms of sample quality (FID) and training efficiency. The ablation study on the importance of initialization further strengthens the authors' claims.

Weaknesses

1.	The existing ablation studies in the paper are already quite convincing; however, including comparative results between the CSD loss function and other score distillation losses would make the work more comprehensive.

2.	Although the appendix provides detailed information, it would be beneficial to include a more high-level flowchart in the main Methods section to summarize the overall two-stage, three-step training process, giving readers a clearer and more intuitive understanding of the general framework.

---

> ### Author Rebuttal · Authors · 2025-07-31
>
> - Thank you for the insightful question! Please see our detailed response below. We’re happy to continue the discussion if any concerns remain.
>
> > ### **Weakness 1: Including comparative results between the CSD loss function and other score distillation losses**
>   - Thank you for your insightful suggestion!
>   - As we mentioned in Sec 3.2.2, our CSD loss is compatible with any valid score distillation loss function (e.g., DMD loss or SiD loss). In our main experiments, we used **SID loss with $\alpha=1.0$** as the default setting.
>   - In our understanding, the reviewer is suggesting we compare against other types of score distillation losses. To address this, we provide additional results below comparing different loss functions. We choose SID loss with varying $\alpha$ values and DMD loss as options. Due to time constraints of the rebuttal period and the limtied computational resources, all experiments are conducted under the default setting with 40,000 guidance iterations and 8,000 generator iterations.
>
>   |Model| SiD alpha=0.8 |SiD alpha=1.0 (default)|SiD alpha=1.2|DMD|
>   |------------|:---------------:|:-------------------------:|:---------------:|:-------:|
>   | VAR-d16 |8.28|8.35|**8.23**|15.70|
>   | LlamaGen-L |- |**7.95**|-|8.13|
>
>   - It can be seen that our method is relatively robust to different values of $\alpha$ in the SiD loss. For DMD loss, while it performs worse on the VAR model, it achieves performance comparable to SiD loss on the LlamaGen model. Overall, we recommend our default setting based on SiD loss with $\alpha=1.0$.
>
> > ### **Weakness 2: It would be beneficial to include a more high-level flowchart in the main Methods section**
>   - Thank you for the helpful suggestion! We agree that including a high-level flowchart would improve the clarity of our presentation.
>   - We have already illustrated the main components of our method in **Fig 5**. However, due to the limitation of space, we were unable to include the multi-stage training pipeline and other implementation details in this figure. We put them in the appendix instead. We will carefully consider your suggestion and revise the figure accordingly in the final version.

---

> > ### Comment · Reviewer_qqvy · 2025-08-04
> > **Response to Authors**
> >
> > The authors have addressed my concerns, and I maintain my stance of accepting the paper. The authors should incorporate the revised content into the final version.

---

> > > ### Author Response · Authors · 2025-08-05
> > > **Thanks for your reply**
> > >
> > > Thank you for your response! We're glad to hear that your concerns have been addressed. We will incorporate the revised content into the final version.

---

### Official Review · Reviewer_cqPi · 2025-06-30

**Clarity:** 2
**Significance:** 3
**Originality:** 2
**Rating:** 4
**Confidence:** 3

**Summary:**

Distilled Decoding 2 (DD2) is a method for collapsing an image autoregressive (AR) model’s multi-step sampling into a single forward pass without a predefined mapping. The key idea is to reinterpret the teacher AR model’s next-token probabilities as conditional score functions in the VQ embedding space, and then to train a one-step generator and  a conditional guidance network that estimates the generator’s own conditional score. Conditional Score Distillation (CSD) loss aligns the teacher and student score functions, minimizing this loss forces the student’s full-sequence distribution to match the teacher’s. The authors also introduce a two-stage initialization, fine-tuning an AR-diffusion model with a Ground-Truth Score (GTS) loss, to stabilize training. On ImageNet-256 with two strong AR backbones (VAR and LlamaGen), DD2 reduces sampling from 10 or 256 steps to one step with minimal FID increase from 3.40 to 5.43.

**Questions:**

see weaknesses, and
1. Why do the student network and the guidance network both use the MAR architecture? Is this necessary, and could the paper’s improvements be due to MAR’s continuous representations?
2. To verify the importance of initialization, the authors conduct diffusion distillation experiments on the ImageNet-64 dataset using the original DMD method. Why did you choose to use DMD for these experiments instead of validating it on CSD?
3. Compared to DD1, DD2 eliminates the pre-defined mapping. However, the benefits of this change are not analyzed in the paper.

**Ethical Concerns:**

["NO or VERY MINOR ethics concerns only"]

**Final Justification:**

I thank the authors for their answers. I suggest that the authors explicitly explain in the final version of the paper why their method is fundamentally different from diffusion score distillation, as well as clarify the advantages of the statement in the abstract, "Compared to DD1, DD2 eliminates the pre-defined mapping." This will help readers better understand the contributions of the paper. Based on this, I'll raise my score.

**Limitations:**

yes

**Quality:**

3

**Strengths And Weaknesses:**

strengths
1. Compared with DD1, DD2 achieves much better one-step sample quality and dramatically faster training.
2. The AR-diffusion fine-tuning with GTS loss is insightful and clearly shown to be crucial for stable, efficient distillation.
3. The paper’s method is evaluated on multiple types of AR models (VAR, LlamaGen), which differ significantly across several key aspects. This demonstrates the effectiveness and generalizability of DD2.

weaknesses
1. The authors emphasize that their method is “fundamentally different from diffusion score distillation, with different goals and challenges from previous work”; however, this distinction is not clearly highlighted in the paper. In my opinion, the most critical aspect of applying score distillation methods to AR models is the definition of the AR model’s score function (or velocity), which has already been addressed in [1]. The subsequent methods presented in the paper seem largely similar to those in diffusion score distillation.
2. The paper uses the GTS loss to fine-tune the teacher model due to structural mismatch. However, this seems to be caused by the use of MAR in the guidance and student model. If MAR is not used and the student model shares the same structure as the teacher model, fine-tuning the teacher model may not be necessary.
3. In Eq (6), It seems that $s_\psi(q_i^{t_i},t_i | q_{<i})$ should be  $s_\Psi(q_i^{t_i},t_i | q_{<i})$. The authors should carefully review the formulas and symbols for possible errors.

[1] Liu, Enshu, et al. "Distilled decoding 1: One-step sampling of image auto-regressive models with flow matching." arXiv preprint arXiv:2412.17153 (2024).

---

> ### Author Rebuttal · Authors · 2025-07-31
>
> - Thank you for the insightful questions! Please see our detailed response below. We’re happy to continue the discussion if any concerns remain.
>
> > ### **Weakness 2 & Question 1: MAR architecture**
>
>    Our response is organized into three sub-questions:
>   1. **Why do the student network and the guidance network both use the MAR architecture?**
>       - The generator must use an MAR-style structure (i.e., using an MLP to output continuous embeddings instead of probability vectors), because a one-step generator must should directly produce continuous samples. Token-wise probabilities (as in AR models) lead to independent sampling at each position and prevent gradient from back propagating through the CSD loss. Thus, a differentiable MLP head is essential.
>       - The guidance model is trained to predict the conditional score of the generator’s distribution, which is also a continuous vector in the codebook embedding space. Therefore, it also requires an MLP architecture similar to MAR to operate in this continuous domain.
>   2. **Is AR-diffusion training necessary?**
>       - As explained above, AR-style models (outputting token probabilities) are unsuitable as generator or guidance networks in our framework. Hence, we cannot use a shared architecture between student and teacher. In this case, AR-diffusion initialization becomes important (see Tab.4), and the proposed GTS loss proves effective (Tab.5).
>   3. **Is the performance gain simply due to using MAR’s continuous representations?**
>       - We believe that the performance improvement is **not brought by the MAR architecture**. DD1 also employs an MLP head that outputs continuous representations like MAR (see Appendix B of [1]) for details). Since DD2 outperforms DD1 with similar architectural, we argue that the improvement stems from the new training strategy, not the continuous representations.
>
>   [1] Liu et al., Distilled Decoding 1: One-step Sampling of Image Auto-regressive Models with Flow Matching.
>
> > ### **Weakness 1: Distinction with diffusion score distillation**
>   - Thank you for your thoughtful question! We would like to take this opportunity to clarify the novelty and contributions of our work.
>   - **Technical Contribution/Novelty**:
>     - We discuss the high level non-trivial differences between our methods and diffusion score distillation & DD1 below:
>       - **Compared to Diffusion Score Distillation**:
>         - The problem we addressed is fundamentally different. Diffusion score distillation focuses on reducing the number of sampling steps in diffusion models. In contrast, our work targets one-step sampling for pre-trained AR models, which is under explored despite AR models achieving competitive or even superior performance compared to diffusion models[2][3][4]. We believe this makes our direction independently meaningful.
>         - To the best of our knowledge, no prior work has directly applied score distillation to AR models. We are the first to enables this application by introducing a way to compute conditional scores with AR models. Accordingly, our guidance network is redesigned from a diffusion-based model to an AR-diffusion model, which brings a non-trivial architectural and conceptual shift.
>         - From a high-level perspective, our method can be seen as decomposing score distillation into a sequence of AR steps. As discussed in Sec. 3.2.2, DD2 can be viewed as applying score distillation to each token position sequentially following the AR order, essentially splitting the original global distillation task into a series of simpler sub-tasks. This token-wise perspective is unique to our approach.
>       - **Compared to DD1**:
>         - While DD1 introduced a way to compute conditional velocities, it mainly used them to construct a static mapping and did not leverage this information in a more principled way. In contrast, DD2 explicitly integrates this information into the training objective, leading to more effective guidance.
>     - **Furthermore, directly applying score distillation to AR models is not feasible without several key adaptations**. Beyond our high-level idea, we have made multiple technical innovations that serve as standalone contributions:
>       - As discussed before, one-step generators must output generated samples directly, rather than probability distributions. Therefore, we replace the classification head of the pretrained AR model with an MLP that outputs continuous embeddings.
>       - We identify the sensitivity of diffusion-style distillation to initialization. To address this, we propose adapting the pre-trained discrete AR model into an AR-diffusion model as an initialization. This modification has proved important (see Tab 4).
>       - We observed that standard AR-diffusion loss is not efficient. Thus, we propose the GTS loss, which trains the AR-diffusion model using ground-truth scores as targets and converges much better (see Tab 5).
>
>   - **Significant Performance Gains**:
>     - As far as we know, DD1 is the only prior work that successfully compresses AR models into one-step generation. DD2 achieves **significantly better performance and lower training cost** compared to DD1, thus pushing the boundary of one-step AR generation by a large margin.
>   - **Broader Impact**:
>     - Our strong results offer a new perspective on training one-step generative models. Previously, the dominant strategy for training such models focused on diffusion-based frameworks. In contrast, our method demonstrates that following the AR schedule is also a highly competitive approach. As shown in the results below (adapted from the Shortcut model paper [5]), our method surpasses several representative and recent diffusion distillation techniques, offering an effective and scalable alternative.
>
>     ||DD2-VAR-d16|DD2-VAR-d20|DD2-VAR-d24|PD|CD|CT|Reflow|Shortcut|
>     |-----|:-------------:|:-------------:|:-------------:|:------:|:-------:|:------:|:--------:|:----------------:|
>     |FID|6.21|5.43|4.91|35.6|136.5|69.7|44.8|10.6|
>
>   [2] Tian et al., Visual Autoregressive Modeling: Scalable Image Generation via Next-Scale Prediction.
>
>   [3] Sun et al., Autoregressive Model Beats Diffusion: Llama for Scalable Image Generation.
>
>   [4] Han et al., Infinity: Scaling Bitwise AutoRegressive Modeling for High-Resolution Image Synthesis.
>
>   [5] Frans et al., One Step Diffusion via Shortcut Models.
>
> > ### **Weakness 3: Wrong symbol**
>   - Thank you for pointing this out! You're right — we should have used a different letter, such as $\Psi$, to represent the AR-diffusion model, since $\psi$ is already used for the guidance network. We will revise all related notations and double-check the formulas in the revision.
>
> > ### **Question 2: Why did you choose to use DMD for these experiments instead of validating it on CSD?**
>   - We have already validated the effectiveness of AR-diffusion initialization under our CSD loss setting in the **ablation studies**. As shown in **Tab 4**, omitting initialization for either the generator or the guidance network leads to performance degradation or even training collapse.
>   - In Sec 3.3, we intentionally use the original DMD setup to demonstrate that the sensitivity to initialization arises from the nature of **score distillation itself**, not from our specific modifications. This strongly motivates our introduction of AR-diffusion tuning as an effective initialization strategy.
>
> > ### **Question 3: The benefits of eliminating pre-defined mapping**
>   - We provide the following intuitive explanation for why removing the pre-defined mapping from DD1 leads to better training:
>     - More effective utilization of knowledge from the pre-trained model:
>  The pre-defined mapping in DD1 provides only an end-to-end training signal. Due to the complexity of the mapping construction process, this signal is difficult for the model to learn. In contrast, DD2 directly aligns the score at each token position, providing a much richer and more localized supervisory signal.
>     - Reduced accumulation of error:
>       - DD1 constructs the mapping autoregressively, where the target token at each position depends not only on the noisy token at that position, but also on previously generated tokens. If the model fails to fit the mapping at one position, the errors will propagate to later positions, leading to cumulative error.
>       - In DD2, the model aligns the conditional distribution of the generator (given previous tokens) with that of the teacher at each position. Thanks to the strong generalization ability of the pretrained teacher model, it can still provide a meaningful conditional distribution even when the previous tokens are imperfect. This behavior is also evidenced by our **teacher-involved sampling results** shown in Table 2. As a result, DD2 suffers from significantly less error accumulation.
>   - In addition, more generally, training generative models without relying on pre-defined mappings allows the model to **discover smoother latent representations** of the target data distribution automatically. **This not only benefits training (because smoother representation is easier to learn), but also improves the model's ability to downstream tasks such as latent interpolation**, which is an important downstream task in generative modeling[6].
>     - To quantify this property, we adopt the **Perceptual Path Length (PPL)** metric proposed in [6], where a lower value indicates smoother interpolation in latent space. We evaluate this metric on both DD2 and DD1.
>
>     ||DD1|DD2|
>     |-----|:---------:|:--------:|
>     |PPL|18437.6|7231.9|
>
>     - The results are shown below, and clearly demonstrate that **DD2 achieves significantly smoother latent interpolation than DD1**. This is very likely because the pre-defined mapping in DD1 itself is not smooth, which inherently limits the smoothness of the learned latent representation.
>
>   [6] Karras et al., A Style-Based Generator Architecture for Generative Adversarial Networks.

---

> ### Comment · Reviewer_cqPi · 2025-08-05
> **discussion**
>
> I thank the authors for their answers.
> I suggest that the authors explicitly explain in the final version of the paper why their method is fundamentally different from diffusion score distillation, as well as clarify the advantages of the statement in the abstract, "Compared to DD1, DD2 eliminates the pre-defined mapping." This will help readers better understand the contributions of the paper.
> Based on this, I'll raise my score.

---

> > ### Author Response · Authors · 2025-08-05
> > **Thanks for your reply**
> >
> > We appreciate your positive feedback and are glad the rebuttal resolved your concerns! We will incorporate the corresponding discussion into the final version.

---

### Official Review · Reviewer_ddiT · 2025-07-01

**Clarity:** 3
**Significance:** 3
**Originality:** 3
**Rating:** 5
**Confidence:** 3

**Summary:**

This paper proposes Distilled Decoding 2 (DD2), a method to accelerate sampling in image autoregressive (AR) models by enabling one-step generation. Unlike Distilled Decoding 1 (DD1), which relies on flow-matching and predefined mappings, DD2 reframes the AR model as a conditional score model. A Conditional Score Distillation (CSD) loss is introduced to match the token-level conditional scores between a pre-trained teacher AR model and a one-step generator. A separate guidance network is trained to approximate the conditional score of the generator, and the generator is optimized to minimize the divergence from the teacher’s score. The method significantly improves both training efficiency (up to 12.3× faster) and generation speed (up to 238× faster inference) with minimal degradation in sample quality on ImageNet-256 using strong AR backbones like VAR and LlamaGen.

**Questions:**

I think this paper is overall good. Please refer to the weakness part.

**Ethical Concerns:**

["NO or VERY MINOR ethics concerns only"]

**Final Justification:**

This paper is overall good for me, so I will keep my rating. I hope the authors could provide more evaluation results, e.g., larger resolution.

**Limitations:**

Yes

**Paper Formatting Concerns:**

The paper is well formatted.

**Quality:**

3

**Strengths And Weaknesses:**

**Strengths**:
1. The idea to view an AR model as a conditional score model and utilize it for training a one-step generator is quite novel and interesting.
2. DD2 shows substantial improvements over DD1, reducing the FID gap between one-step and teacher AR models.
3. Experiments span multiple strong AR backbones (VAR, LlamaGen) and analyze one-step and multi-step performance with meaningful baselines. A careful study of initialization reveals its importance in training stability, with a tailored strategy to reuse teacher model knowledge.

**Weakness**:
1. Only ImageNet-256 is used for evaluation. It would be better to test on larger resolutions and more diverse datasets, or conditional generation settings like text-to-image.

---

> ### Author Rebuttal · Authors · 2025-07-31
>
> - Thank you for your positive feedback! Please see our detailed response below. We’re happy to continue the discussion if any concerns remain.
>
> > ### **Weakness 1: Lack of large-scale experiments**
>   - Thank you for the suggestion! We agree that larger-scale experiments would further strengthen the paper.
>   - Due to the limited time and computational resources available during the rebuttal period, we are unable to conduct larger-scale experiments at this time. However, we plan to include them in the revision and future work.
>   - Additionally, we would like to offer the following clarification to address your concern: ImageNet-256 is a sufficiently large and diverse dataset, and generation on this dataset is considered challenging. It has been widely adopted as the main benchmark in many recent works on new generative paradigms [1][2][3][4][5]. Our choice of dataset follows the setting in DD1. Therefore, we believe that using ImageNet 256 provides a meaningful and representative evaluation of our method.
>
>   [1] Tian et al., Visual Autoregressive Modeling: Scalable Image Generation via Next-Scale Prediction.
>
>   [2] Sun et al., Autoregressive Model Beats Diffusion: Llama for Scalable Image Generation.
>
>   [3] Frans et al., One Step Diffusion via Shortcut Models.
>
>   [4] Liu et al., Distilled Decoding 1: One-step Sampling of Image Auto-regressive Models with Flow Matching.
>
>   [5] Geng et al., Mean Flows for One-step Generative Modeling.

---

### Official Review · Reviewer_A7Mc · 2025-07-03

**Clarity:** 3
**Significance:** 2
**Originality:** 2
**Rating:** 4
**Confidence:** 4

**Summary:**

The paper introduces Distilled Decoding 2 (DD2), which enables one-step sampling in image autoregressive models, addressing the slow generation speeds limitation in traditional token-by-token AR sampling. Unlike prior work such as DD1, which relies on learning an explicit mapping via flow matching, DD2 reframes the AR model as a conditional score model predicting gradients of log-probabilities in the latent embedding space. It proposes a Conditional Score Distillation loss that trains a one-step generator and a guidance network to match the conditional scores of the original AR model, ensuring the output distribution remains faithful to the teacher model without predefined mappings.

**Questions:**

1. For an autoregressive model, the entropy is usually higher at the beginning of the sentence and lower later in the sentence, which makes the scores of later tokens harder to model than those of the initial tokens. I am wondering whether any weighting scheme can be used for tokens in different positions. A simple choice is to normalize by the norm of the score function across the sequence length. This would be different from the score normalization used in SID (across diffusion time).

2. Since you have access to the ground truth score of the teacher model, why not use that in equation 5 (like the loss of equation 6)? It should be more sample-efficient than denoising score matching loss.

**Ethical Concerns:**

["NO or VERY MINOR ethics concerns only"]

**Final Justification:**

My conerns are all resolved. I will keep my score. Hope the generation results can be furthur improved.

**Limitations:**

1. The quality of the one-step generation can still be improved.
2. The advantage compared to the one-step diffusion model (distilled from a diffusion model) is not very clear.

**Quality:**

3

**Strengths And Weaknesses:**

The quality and clarity are good. I commented other pros and cons below.

Strengths:

1. Usually, it is difficult to perform distillation from a clean, non-diffusion teacher model. In this case, since the teacher is a categorical distribution whose Gaussian convolution has a closed form (as used by DD1), this is a smart choice. It avoids the need for teacher score estimation, which is often required when the target distribution is an intractable EBM (e.g., [1]).

2. The proposed conditional score distillation method is straightforward and natural.

3. The most challenging part of this type of distillation research is that the student needs to be initialized from the teacher model, which is not straightforward when the architectures of the teacher and the student differ. It is great that the authors conducted a detailed study of this problem. The proposed method is heuristic but intuitive.

Weaknesses:
See the questions section, especially question 2.

---

> ### Author Rebuttal · Authors · 2025-07-31
>
> - Thank you for the insightful questions! Please see our detailed response below. We’re happy to continue the discussion if any concerns remain.
>
> > ### **Weakness 1: Whether weighting schemes can be applied to different token positions**
>   - Thank you again for raising this insightful point!
>   - In our default setting, we adopt a weighting scheme similar to SID[1], where each data sample is weighted by the norm of its score function. We use different weights for different data samples, which is different from your suggestion, where different weights are applied to each token position. To investigate this idea, we experimented with your proposed approach, where each token position is weighted by the norm of its score function.
>   - In addition, to further explore the impact of token-position-dependent weighting schemes on training, we also explored two static, token-wise weighting strategies:
>     - The weights linearly decrease from 3 to 1 across token positions from the first to the last.
>     - The weights linearly increase from 1 to 3 across token positions from the first to the last.
>   - We conducted experiments using the VAR-d16 model. Due to time constraints of the rebuttal period and limtied computational resources, we trained under the default setting with only 40,000 guidance iterations and 8,000 generator iterations. The results are summarized below:
> | Schedule | Default | Token Wise Norm | Static from 3 to 1 | Static from 1 to 3 |
> |:----------:|:---------:|:-----------------:|:--------------------:|:--------------------:|
> | FID      | 8.35    | 10.85           | **6.91**           | 10.44              |
>   - The results show that adjusting the token-wise loss weighting schedule is indeed beneficial. Assigning higher weights to tokens at the beginning of the sequence leads to much better performance compared to our default setting.
>     - **Possible reason**: During training, the generator's objective at each token position is to align the conditional distribution of that token (given the previously generated tokens) with the corresponding distribution provided by the pretrained autoregressive model. This implies that tokens earlier in the sequence are responsible for providing accurate conditioning for subsequent tokens. If these early tokens are not well-trained, the learning targets for later tokens may also become unreliable. Therefore, increasing the loss weight for early tokens can help improve their learning quality and, in turn, enhance overall training effectiveness.
>
>   [1] Zhou et al., Score identity Distillation: Exponentially Fast Distillation of Pretrained Diffusion Models for One-Step Generation.
>
> > ### **Weakness 2: Why not use Eq. (5) to train the conditional guidance network?**
>   - Thank you for your careful reading!
>   - The loss in Eq. (6) is used to train a model to output the conditional score of the **teacher (i.e., the pre-trained AR model) distribution**. It relies on the pretrained AR model to provide the **ground truth conditional probability** of each token given all its previous tokens. This allows us to explicitly compute the ground-truth score for each token.
>   - In contrast, Eq. (5) defines the training objective for the guidance network, which aims to learn the conditional score of the **one-step generator distribution**. Since the one-step generator directly **outputs the generated sample** without explicitly modeling the conditional distribution of each token given the previous ones as the pretrained AR model does, we are unable to compute the exact ground-truth score for the generator.
>   - As a result, we follow the common practice in traditional diffusion and AR-diffusion frameworks and train the guidance network by approximating the score function using a Monte Carlo estimatation for each token's score, as formulated in Eq. (5).
>
> > ### **Limitation 1: The quality of the one-step generation can still be improved.**
>   - This is because our training process relies solely on the information provided by the pretrained AR model. Thus, the performance is inherently bounded by that model. Nevertheless, DD2 has already achieved a relatively small performance loss compraed to the pretrained AR model. We believe that incorporating real samples in the training process with adversarial objectives (e.g., GAN loss as used in [2][3]) could further overcome this limitation, which we leave for future work.
>
>   [2] Kim et al., Consistency Trajectory Models: Learning Probability Flow ODE Trajectory of Diffusion.
>
>   [3] Yin et al., Improved Distribution Matching Distillation for Fast Image Synthesis.
>
> > ### **Limitation 2: The advantage compared to the one-step diffusion model is not clear.**
>   - We compared several representative and recent methods in diffusion distillation on the ImageNet 256×256 benchmark, as shown below. Results of diffusion-based are taken from [4]. As can be seen, DD2 achieves significantly better performance compared to these methods.
>
>   |     | DD2-VAR-d16 (ours) | DD2-VAR-d20 (ours) | DD2-VAR-d24 (ours) | PD[5] | CD[6] | CT[6] | Reflow[7] | Shortcut Model[4] |
>   |-----|:--------------------:|:--------------------:|:--------------------:|:-------:|:-------:|:-------:|:-----------:|:-------------------:|
>   | FID |  6.21               | 5.43               | 4.91               | 35.6  | 136.5 | 69.7  | 44.8      | 10.6              |
>    - We would like to emphasize that the problem we aim to address is **fundamentally different from diffusion distillation**. Specifically, we focus on pre-trained AR models, while diffusion distillation is based on pre-trained diffusion models, which is a fundamentally different type of generative models. As a result, existing diffusion distillation methods cannot be directly applied to AR models.
>
>   - Notably, AR models have already achieved performance comparable to, or even surpassing, that of diffusion models [8][9][10]. Therefore, addressing the distillation of AR models presents an independently meaningful and valuable research direction.
>
>    [4] Frans et al., One Step Diffusion via Shortcut Models.
>
>    [5] Salimans & Ho, Progressive Distillation for Fast Sampling of Diffusion Models.
>
>    [6] Song et al., Consistency Models.
>
>    [7] Liu et al., Flow Straight and Fast: Learning to Generate and Transfer Data with Rectified Flow.
>
>    [8] Tian et al., Visual Autoregressive Modeling: Scalable Image Generation via Next-Scale Prediction.
>
>    [9] Sun et al., Autoregressive Model Beats Diffusion: Llama for Scalable Image Generation.
>
>    [10] Han et al., Infinity: Scaling Bitwise AutoRegressive Modeling for High-Resolution Image Synthesis.

---

> > ### Comment · Reviewer_A7Mc · 2025-08-04
> > **Thanks for your reply**
> >
> > Thank you for the author's reply.
> >
> > There is also a recent paper that discusses how to combine VSD with training data [1], which might be useful for improving generation quality. I have no further concerns.
> >
> > [1] Towards Training One-Step Diffusion Models Without Distillation

---

> > > ### Author Response · Authors · 2025-08-05
> > > **Thanks for your reply**
> > >
> > > Thank you for your response! We're pleased to hear that our rebuttal has addressed your concerns.
> > >
> > > We also appreciate your recommendation of paper [1], which introduces a new approach for performing diffusion score distillation without relying on teacher scores. Moreover, [1] provides an insightful discussion on the importance of teacher initialization, which supports and complements our analysis in Section 3.3. We will include a discussion of this work in our final revision.
> > >
> > > [1] Zhang et al., Towards Training One-Step Diffusion Models Without Distillation

---

### Note · Authors · 2025-08-12

We thank the Reviewers and the AC for their constructive discussions and time. We also appreciate all the reviewers for their unanimous positive feedback, with the final scores of 4, 5, 4, and 4 as discussed. Below we reiterate the key points:
- **Main contributions of our paper:**
  - **Novel methodology:** We propose conditional score distillation, introducing score distillation (SD) into a new problem of one-step compression for **AR models**. providing a new method for training one-step generative models.
  - **Technical innovations:**
    - Architectural modifications for continuous outputs from the one-step generator and conditional score model.
    - AR-diffusion tuning for initialization.
    - Using GT scores to improve initialization efficiency.
  - **Impressive results:**
    - Compared to DD1, our method achieves substantial improvements in both performance (reduce the gap between one-step model and teacher model by up to 67%) and training time (up to 12.3x speedup). To the best of our knowledge, DD1 is the only prior work enabling one-step sampling for image AR models, and DD2 marks a significant advancement in this direction.
    - DD2 also significantly outperforms typical diffusion distillation methods in the one-step setting.
  - **Potential broader impact:** Our results suggest AR formulation can be a promising alternative to diffusion formulation for one-step model training, offering a new perspective.
- **Additional experiments during the rebuttal phase:**
  - **Mapping smoothness (cqPi):** DD2 doesn't rely on pre-defined mappings, allowing it to discover better mappings. We verified that DD2 yields much smoother mappings than DD1 (PPL 7231.9 v.s. 18437.6), making it easier to learn and better suited for tasks like latent interpolation.
  - **AR-wise loss schedule (A7Mc):** We identified a better loss weight schedule, which assigns larger weights to earlier positions, suggesting room for further improvements.
  - **Loss type ablation (qqvy):** We found different types of SD losses all work for DD2.
- **Planned updates:**
We greatly appreciate the Reviewers’ feedback on writing. We will revise the final version accordingly, including:
  - Discuss differences between DD2 and diffusion score distillation (cqPi).
  - Discuss benefits of DD2’s mapping-free property (cqPi).
  - Include a workflow flowchart (qqvy).
  - Expand related work discussion (A7Mc).

  We will also try to apply DD2 to text-to-image tasks in the future (ddiT).

---

### Decision · Program_Chairs · 2025-09-17

**Decision:**

Accept (poster)

**Comment:**

This paper proposes a method for one-step sampling of autoregressive models using conditional score distillation. The approach is intuitive, efficient, and well-validated on ImageNet-256, showing large gains in both training and inference speed while preserving competitive quality. Reviewers were split initially, but concerns about novelty and clarity were largely addressed during rebuttal, leading to stronger support. Overall, the work is timely, impactful, and I recommend acceptance.